# City Branding and Industrial Transformation from Manufacturing to Services: Which Pathways do Cities in Central China Follow?

**Meiling Han [1], Martin de Jong [2,3] and Minghui Jiang [1,*]**

1   School of Management, Harbin Institute of Technology, Harbin 150000, China; Meilinghanalice@gmail.com
2   Rotterdam School of Management & Erasmus School of Law, Erasmus University Rotterdam, 3000 Rotterdam, The Netherlands; W.M.Jong@law.eur.nl
3   School of International Relations and Public Affairs, Fudan University, Xiamen 361000, China
*   Correspondence: jiangminghui@hit.edu.cn; Tel.: +86-138-3614-5212

**Abstract:** A potentially attractive way for cities to maintain economic growth while reducing environmental harm is to let their production structures undergo industrial transformation, a process otherwise known as ecological modernization. This attraction lies mainly in the fact that residents, visitors and corporations prefer clean air, water and soil as a milieu to invest their resources in. Municipal governments can use city branding as an important instrument to force off such a transformation, if it is taken as a point of departure for the adoption of a strategy to which they are deeply committed and for the benefit of which they are willing to deploy their various policy instruments. In the literature on ecological modernization, five different pathways for industrial transformation in cities have been identified and these have been matched with city branding practices. In this contribution, the abovementioned conceptual framework is further detailed and specified to account for a variety in types of secondary and tertiary sector industries. In the empirical sections, all cities in the Chinese provinces Hubei and Hunan, where the transition from manufacturing to services is typically most pressing, are examined in terms of their industrial structures, pathways to industrial transformation and city branding choices. The results indicate, inter alia, that further subdivision of the secondary and tertiary economic sectors is useful in understanding key features of the transformation, and that different sub-pathways affect tradeoffs between economic expansion and ecological preservation differently. Branding practices among Hubei and Hunan cities also indicate that some industries are more easily embraced and utilized than others in establishing brand identities and adopting popular city labels.

**Keywords:** industrial transformation; industrial structure; pathways; city branding; central China (Hubei and Hunan)

## 1. Introduction

Facing low air quality in their immediate physical environment and global warming as an important background factor, many cities around the world aim to modernize their industrial structures [1]. This is especially true in areas where heavy manufacturing is still dominant, but where hope lives that a transition towards the tertiary sector will be a boost to both Gross Domestic Product (GDP) growth and scores on environmental indicators [2]. Such urban and industrial transformation leading towards 'ecological modernization' is not an easy challenge to deal with, however. The objective underlying ecological modernization is to reduce the use of raw materials and natural resources as well the emission of harmful substances in the production process. This require technological as well

organizational innovation leading to win-win situations where increase in economic value added is paired with decrease in harm done to the natural environment [3,4]. Moreover, in comparison with remedial measures for fixing damage done to the natural environment, ecological modernization has the advantage of emphasizing the principle of pollution prevention: cost saving goes hand in glove with preserving or even improving the ecological environment [5–7]. If economic growth is paired with an absolute decrease in environmentally harmful emission or use of resources, this has been defined as 'absolute decoupling'. If an increase of GDP comes with a less than proportional rise in environmental damage done, this is known as 'relative decoupling' [8]. Although the former is obviously the more desirable option, it is far more common for cities and their industries to realize the latter.

Although it is attractive for cities to realize industrial transformation as ecological modernization, this is neither easy nor painless. Many of them, especially in developing and newly emerging economies, still rely on polluting industries since these generate essential employment and income for a large share of their inhabitants and can only be moved out at high costs. New 'high-quality' visitors, residents and investors with higher levels of professional education, expertise in clean technologies or significant financial resources to set up their corporations prefer clean air, water and soil as a milieu to place their stakes in. For municipal governments, city branding can be an important instrument to attract these desirable new residents and force off such a transformation. This is only true, however, if this brand is taken as a point of departure for the adoption of a broader transformation strategy to which they are deeply committed and for the benefit of which they are willing to deploy various policy instruments as well. City branding has been defined as "the means both for achieving competitive advantage in order to increase inward investments and tourism, and also for achieving community development, reinforcing local identity and the identification of the citizens with their city and activating all social forces to support development efforts" [9]. Dinnie has proposed that an understanding of the interest in city brands may be that all places can benefit from implementing a coherent strategy for managing their resources, reputation and image [10]. In recent years, the interdisciplinary research on city branding has grown [11]. Especially the environmental dimension has obtained a fair amount of attention in the literature [12]. In this study, we bring the processes of ecological modernization and city branding together in cities where urban transformation is felt the strongest. Adopting a powerful city brand can be a crucial trigger for realizing and deepening industrial transformation towards ecological modernization—but not essential. It will only have this effect if operated in line with a broader strategy to implement it, or else it will ring more in the manner of greenwashing. Coherence between adopted city brands and industrial patterns is therefore a key condition for making this happen.

In this contribution, a generic conceptual framework, which includes five developmental pathways for industrial transformation will be introduced, after which it will be further detailed to account for a variety in types of secondary and tertiary sector industries. These more specific sub-pathways throw a more precise light on the transformational trajectories that cities adopt and the ways in which they brand themselves internally and externally. We will examine whether a more detailed understanding of these industrial profiles and existing branding practices among municipalities can help us understand why certain manufacturing and/or service-oriented profiles are emphasized and hailed or rejected a dissimulated in cities' branding strategies. In the empirical sections of this article, all cities in the Chinese provinces Hubei and Hunan, where the transition from manufacturing to services is typically most pressing, are examined in terms of their industrial structures, pathways to industrial transformation and city branding choices.

The remainder of this article will proceed as follows. Section 2 will introduce the conceptual framework for developmental pathways and subsequently specify various sub-pathways. Section 3 will clarify the various methodological steps taken in this study for data collection and analysis. Section 4 will present relevant economic and geographic overview data on the cities in Hunan and Hubei Province later used in the analysis. The result will be a characterization of all cities in terms of pathways and sub-pathways that apply to them. Section 5 will then compare these sub-pathways with the branding choices made by those cities and analyze the coherence between pathways and chosen

brands. Section 6 wraps up this article with conclusions and reflects on the broader research and policy relevance of the approach proposed here.

## 2. Ecological Modernization and a Revised Conceptual Framework on Developmental Pathways

Ecological modernization theory (EMT) has emerged as a prominent neoliberal theory and it is currently one of the leading theories in environmental sociology [13]. Most generally, "the aim of EMT has been to analyze how contemporary industrialized societies deal with environmental crises" [14]. Hajer claim that the central idea of ecological modernization is the growing compatibility between environmental protection and economic growth [15]. Previous studies in ecological modernization all indicate that environment is highly related to industrial production; it was therefore normally adopted as a dependent variable affected by industrial production features [16]. Bicknell's input-output model to calculate the ecological footprint was adopted by Liu Jianxing and Cao Shuyan et al. [17–19] to discuss the relationship between the activities of the three different economic sectors (primary, secondary and tertiary) and their corresponding amounts of pollution. Their results suggest that among the three industries, the secondary industry had a huge impact on the environment and was responsible for a serious imbalance between the supply and demand of ecological resources. The total ecological footprint of the tertiary industry was the smallest, and its land use had the highest economic efficiency, which was conducive to reducing the ecological impact of China's socio-economic metabolism. In more recent academic literature, in which industrial transformation and ecological modernization are connected with city branding practices, a typology of five different developmental pathways for industrial transformation in cities was developed. The types are based on the positions of cities in the urban hierarchy (international, national or regional orientation) and their stage of industrial development (agriculture and extraction oriented, manufacturing oriented, and trade and service oriented) and express expected trajectories for economic development based on the need for ecological modernization. Table 1 below depicts these five pathways as they have been coined in de Jong et al. (2018) and empirically validated for the three Chinese Mega City Regions around Beijing, Shanghai and Hong Kong/Shenzhen/Guangzhou and China's Northeastern provinces Heilongjiang, Jilin and Liaoning in Han et al. (2018) [20,21].

**Table 1.** Ecological modernization and urban developmental pathways.

| Stage of Economic Development/Position within the Region | Primary Sector Dominates | Secondary Sector Dominates | Tertiary Sector Dominates |
|---|---|---|---|
| Regional orientation | PATHWAY 1 Eco-tourism (accommodating manufacturing) | PATHWAY 2 Advanced, low carbon manufacturing | PATHWAY 4 Knowledge and culture-oriented service |
| National orientation | N/A | PATHWAY 2 Advanced, low carbon manufacturing | PATHWAY 4 Knowledge and culture-oriented service |
| International orientation | N/A | PATHWAY 3 High-tech innovation | PATHWAY 5 Global advanced producer services |

In both of these studies, expected developmental pathways of cities as outlined above were established based on various economic and geographic figures and indicators and then compared with the city branding choices that these cities have adopted. Cities on pathways 2 and 4 appeared to be by far most numerous. Cities on pathways 4 and 5 tended to be pleased with their service-oriented profile and their branding strategies tended to be in line with what could be expected from their industrial profile and pathway. Cities on pathway 2 (and 3) often had more mixed emotions about their manufacturing dominated image and either presented themselves as 4 (or 5) or blended their own pathway 2 with features of pathway 4 (or less often, pathway 1). Cities on pathway 1 were comparatively rare. Those that existed branded themselves as blends of 1/2 or 1/2/4. Although these studies were useful in characterizing the challenges municipal governments face in their ecological modernization-inspired self-reinvention, these five pathways proved to be insufficiently specific to

understand more subtle aspects in the tradeoffs made between economic and ecological considerations in city branding and urban transformation. Some types of secondary sector industries in fact were less environmentally harmful than others, while some tertiary sector activities were more economically desirable than others without the framework being able to clarify how this worked. In spite of the doubtful reputation of the secondary sector as being environmentally harmful, demand for its products and employment remains high. Meanwhile, in spite of the intuitive of appeal service industries as a whole have, some of them may actually generate very low value added. Indeed, the secondary and tertiary sectors are both broad concepts including many specific subordinate industries—not all of which have the same economic and ecological impact.

In this study, we aim to become more precise on the existence of various developmental pathways and seek ways to subdivide the three economic sectors on which developmental pathways are based (primary, secondary and tertiary) into more detailed subcategories allowing for a better grasp of their respective economic and ecological attractiveness. Since the primary sector accounts for a low percentage of industrial activity, is only dominant in a very tiny number of cities, is simple and straightforward (including only agriculture, forestry, animal husbandry and fishery) and has limited impact on the ecological environment [22], we decided to leave it intact for the purpose of our analysis. Pathways 2 and 3, in both of which the secondary sector prevails, tell a different story. According to the Chinese National Bureau of Statistics [23], the secondary sector can be subdivided into four main categories (manufacturing, construction, mining and production and distribution of electricity, gas and water), which correspond with pathways 2L/2H or 3L/3H for manufacturing, and 2bcd or 3bcd for the other three categories, as shown in Table 2. In the rest of our analysis, we will not be using all of these subtypes, however. Our empirical data (see Appendix A) show that regarding the secondary sector, only manufacturing industries play a significant role in the city branding practices undertaken by municipalities, with light and heavy manufacturing often explicitly mentioned and the former valued more highly than the latter (see Appendix B). Construction, mining and production of utilities were largely ignored in communication and strategy development, partly because they are common to all cities and therefore have no distinctive value (construction and production of utilities) and partly because they are considered highly extractive and polluting (construction and mining). We will therefore just include light (L) and heavy (H) manufacturing industry from hereon.

**Table 2.** Categorization of the secondary sector and corresponding developmental pathways.

| Main Types of Industries in the Secondary Sector | Subtypes in the Secondary Sector | Corresponding Developmental Pathways |
| --- | --- | --- |
| Manufacturing | Light manufacturing (L), heavy manufacturing (H) | Pathways 2L/3L, Pathways 2H/3H |
| Construction | Housing; construction; construction and installation; architectural decoration and other construction | Pathway 2b/3b |
| Mining | Coal mining and washing; oil and gas extraction; Ferrous metal mining and dressing; non-ferrous metal mining and dressing; non-metallic mining and dressing; other mining | Pathway 2c/3c |
| Production and distribution of electricity, gas and water | Electricity, heat production and supply; gas production and supply; water production and supply | Pathway 2d/3d |

Singelmann and Browning (1978) have analyzed the service-oriented tertiary sector using a so-called six-sector model, in which it was categorized into four broad subtypes based on the nature and objects of the services provided: distributive services (trade, transport, and communication); personal services (hotels, catering, entertainment and miscellaneous personal services); producer services (banking, insurance, business services); and social services (government, health, education, non-profit organizations) [24]. Following Yan [25] who modified this typology, which is also used

in the Standard Industrial Classification released by the Chinese National Bureau of Statistics, we eventually made a classification into producer services, distributive services, consumer services and social services. Producer services are defined as services provided for producers and individuals at intermediate stages [25]. Distributive services relate to the provision of transportation services to the final consumer. Consumer services target the delivery of personal services to the individual consumer. Finally, social services consist of activities aimed at the reproduction of labor (such as health services, education and social welfare) and activities to stabilize social relations between various societal groups and classes (public management). These classifications for the tertiary sector and their corresponding pathways are shown in Table 3.

**Table 3.** Categorization of the tertiary sector and corresponding developmental pathways.

| Main Types of Industries in the Tertiary Sector | Subtypes in the Tertiary Sector | Corresponding Developmental Pathways |
| --- | --- | --- |
| Producer services | Information transmission; computer services and software; financial intermediation; real estate; leasing and business services | Pathways 4P/5P |
| Distributive services | Wholesale and retail trade; traffic, transport, storage and post | Pathways 4D/5D |
| Consumer services | Hotels and catering; service to households and other services; culture, sports and entertainment | Pathways 4C/5C |
| Social services | Scientific research, technical services and geological prospecting; management of water conservation and the environment; education; health, social security and social welfare; public management and social organization; international organization | Pathways 4S/5S |

It should be noted that tourism is not in any of the four categories mentioned above. According to Charles (2011), tourism is a combination of economic activities, services or industries that provide tourists with an out-of-home experience, and this includes transportation, accommodation, catering, retail, entertainment facilities and other hospitality services [26,27]. These can be found dispersed across the various other types. Since we have no a priori reason to believe that any of the four main types of service industries appear either dominant over or more economically or ecologically attractive than the others, we have included all of them in our analysis. That said, at first sight some of the producer services match aspects of the advanced producer services that tend to set global cities apart, while scientific research and technical services are widely assumed to promote innovation. Both of these can be expected to play a bigger role than other types of services in the city branding around ecological modernization [28].

In Section 3 on methodology, we will clarify how we established economic pathways and sub-pathways for all cities in the Chinese provinces of Hunan and Hubei, where the transformation from secondary to tertiary sector dominated industries occurs at its most conspicuous in the country provinces. Since 2013, the GDP proportion of the tertiary sector exceeded that of secondary for the first time in China and the cities in these two provinces are typically in the middle of this transition. This is the reason why we have taken them for our empirical analysis.

## 3. Methodological Steps Taken for Data Collection and Analysis

To establish a city's developmental pathway and sub-pathway, relate it to its branding choices and then examining how tradeoffs are made between attractiveness of economic value added and ecological harm reduction, a certain number of methodological steps were used. They were inspired

by Jong et al. (2018) and Han et al. (2018), but have been amended given the specific focus of this contribution on sub-pathways [20,21].

1. *Sample of cities.* To assess validity and usefulness of the modified developmental pathway framework, we selected all cities in the Chinese provinces Hubei and Hunan. They are both located in the central part of China and constitute a zone bridging the developed eastern coastal provinces with the less developed western inland provinces [29,30] and display many of the urban and industrial transformation features under study here: for long they have had a dominant manufacturing industry, but an obvious transition towards more and more service industries is underway. Yet, cities in these two provinces are still plagued by severe environmental contamination from which local governments can no longer look away [31,32].

2. *Administrative position.* In establishing the position of a city in the administrative hierarchy, we found no international cities in either province. Wuhan and Changsha are capital cities categorized as national cities (NAT). All other cities all were classified as regional (REG). See Table 4. This implies that we will only find cities on pathways 1, 2 and 4.

3. *Developmental stage and dominant sub-sectors.* To establish the developmental (sub) stage of a city, a selection of relevant statistical data was collected. We mapped the percentage of (1) heavy manufacturing H and (2) light manufacturing L in total GDP; data to establish them as percentages of the working population could not be retrieved. Producer services (P), distributive services (D), consumer services (C) and social services (S) were mapped as percentages of GDP and alternatively as a share of the working population if GDP data proved to be missing. Based on these percentages, the dominant subtype(s) of industry could be established and consequently the developmental stage of each city. Again, see Table 4.

4. *Developmental pathway and sub-pathway.* To establish the expected developmental pathway and sub-pathway of each city based on a city's industrial features, the scores from steps 2 and 3 were combined (see Table 4). Take Wuhan for instance: in light of its ratio of (1) working population and (2) GDP to primary, secondary and tertiary economic sectors, which are respectively 9/38/53 and 3.3/43.9/52.8, the service economy is clearly the dominant one. In addition, it is of national rather than regional importance. Therefore, Wuhan's expected pathway is 4. Regarding the sub-pathways, P/D/C/S as GDP of tertiary sector (in percentages) is 32.75/10.94/14.87/41.44, implying that social services prevail. Wuhan is therefore unambiguously on pathway 4S. Meanwhile for Shiyan, the proportion 1/2/3 as percentage of working population is 41.4/18.1/40.6, with only a tiny gap between the primary and tertiary sectors. To add to the complexity, the GDP percentages generated in the three sectors are at 12.1/47.7/40.2, respectively: the secondary industry is dominant on that criterion. We therefore establish the main pathway of Shiyan as 1/4/2. Regarding the sub-pathways, P/D/C/S as GDP of the tertiary sector (in percentages) is at 13.29/39.89/7.45/39.37, indicating that distributive and social services dominate; as for secondary sector, the L/H (light and heavy manufacturing) percentages are 66.9/33.1, showing a clear dominance of light manufacturing. Consequently, Shiyan displays characteristics of pathways 1, 4D, 4S and 2L. This result can be observed in Table 4 in the columns to the right.

5. *City branding choices I.* To measure city branding choice, we used two different indicators: city brand identity and city label. A city brand identity reflects how a city defines itself when comparing itself with other cities [33]. Different from city brand identity, a city label is a generic label that a city uses when promoting itself. Such labels are normally policy-related academic phrases, and easy to remember [34]. Both city brand identities and city labels reflect, in different ways, the developmental pathway a city itself believes it is on or should be on and what it adopts in its internal and external communication and strategy development. City brand identities were identified in self-descriptions of the city in official government documents. These could be Urban Master Plan, 12th Five Year Social and Economic Plan or 13th Five Year Social and Economic Plan. These could only be taken from central parts of these documents such as summaries, introductions or conclusions, since city brand identities should be found at conspicuous places of documents.

Often the interpretation of one or a few such key phrases and relating these to adopted pathway and sub-pathway required a fair amount of interpretation but was based on terms bold-faced in the text. See Table 5.

6. *City branding choices II.* To establish which city labels were most popular, these were counted in the same planning documents mentioned under 5. However, in this case, not one central key phrase was used as for the city brand identity, but a total count of each city label was made in each plan document. In this manner, the frequency of their occurrence across the plans allowed us to calculate a second indicator for the adopted pathway of a city. The city labels for the categorization of the main pathways are shown in Appendix E. For instance, 'service center', 'trade center', 'financial center' feature under service city, and service city belongs to pathways 4 and 5. Of these labels we calculated the frequencies, the results of which are shown in Appendix A. For instance, in Appendix A we can see that for Wuhan and Shiyan, the frequency of the city label 'service city' very evidently occupies the first place in the three documents. Therefore, Wuhan's adopted pathway is 4 (see Appendix A); For Huangshi, 'service city' and 'advanced manufacturing city' are the two most frequently mentioned city labels, so the pathway adopted is 4/2. In order to determine the adopted sub-pathways, another frequency count in those same official documents was made for each city, this time to map how many times each of the various subtypes of industries under step 3 was mentioned. Dominance of such a subtype allowed us to establish the adopted sub-pathway in each city. See Appendix B for the secondary economic sector and Appendix C for the tertiary sector. If we take Wuhan as an example, we see that its sub-pathway is 2H (Appendix B) and 4D (Appendix C). As a procedure, we first determine its main pathway: pathway 4. Secondly, we find the corresponding sub-pathway, which is 4D. Likewise, in the case of Huangshi, its main pathways are 4/2, and its sub-pathways 4D/2H.

7. *Matching adopted with expected pathways.* Finally, to examine to what extent the city brand identities and city labels expressed in adopted sub-pathways match the expected sub-pathways given their economic and geographic data, we combined the expected pathways derived under step 4 with the adopted pathways from step 6 (main pathways) and from step 7 (sub-pathways) into Appendix A. Where expected, sub-pathways and adopted sub-pathways significantly differed from each other, which was flagged up for subsequent discussion and could be used to understand how cities make tradeoffs between economic and ecological elements in choosing an industrial profile and developmental pathway.

## 4. The Provinces of Hunan and Hubei Province and their Cities

### 4.1. Hunan Province

Located in China's central region (see Figure 1), Hunan is not only the heart of its agricultural production, but also its main distribution center for industrial products, including steel, machinery and electronics. In 2016, the GDP proportions for the primary, secondary and tertiary sectors were 9.4%, 43.2% and 47.4%, compared to 8.8%, 41.7% and 49.5% just a year later. The typical of industrial readjustment towards services is therefore in full process in Hunan. The development of the tertiary sector including Finance, Real estate, Scientific research, Comprehensive technical supporting services, Public management and Social services [35], is promising from both an economic and ecological viewpoint [36,37].

Hunan province counts 13 cities, including: Changsha, Zhuzhou, Xiangtan, Hengyang, Shaoyang, Yueyang, Changde, Zhangjiajie, Yiyang, Loudi, Chenzhou, Yongzhou, Huaihua. Changsha is the provincial capital and an important high-speed railway and aviation hub and important industrial and commercial center in the central region. In 2016, Changsha's GDP reached 932.37 billion yuan, an increase of 9.4% over that of 2015. Changsha ranks 13th in China and second in China's central region. Zhuzhou and Hengyang benefit from their geographical location and have become its logistics center and old industrial base. Yueyang is an ancient international trading port city in China. Changde is a sub

provincial center city in Hunan and also a city where light manufacturing is especially strong. Shaoyang, Yiyang and Loudi are all secondary sector oriented, with manufacturing occupying more than half of the total output. Chenzhou, Yongzhou, Huaihua and Zhangjiajie are rich in forest resources; Zhangjiajie is also a major tourist city. Xiangtan is known nationwide as being Mao Zedong's hometown.

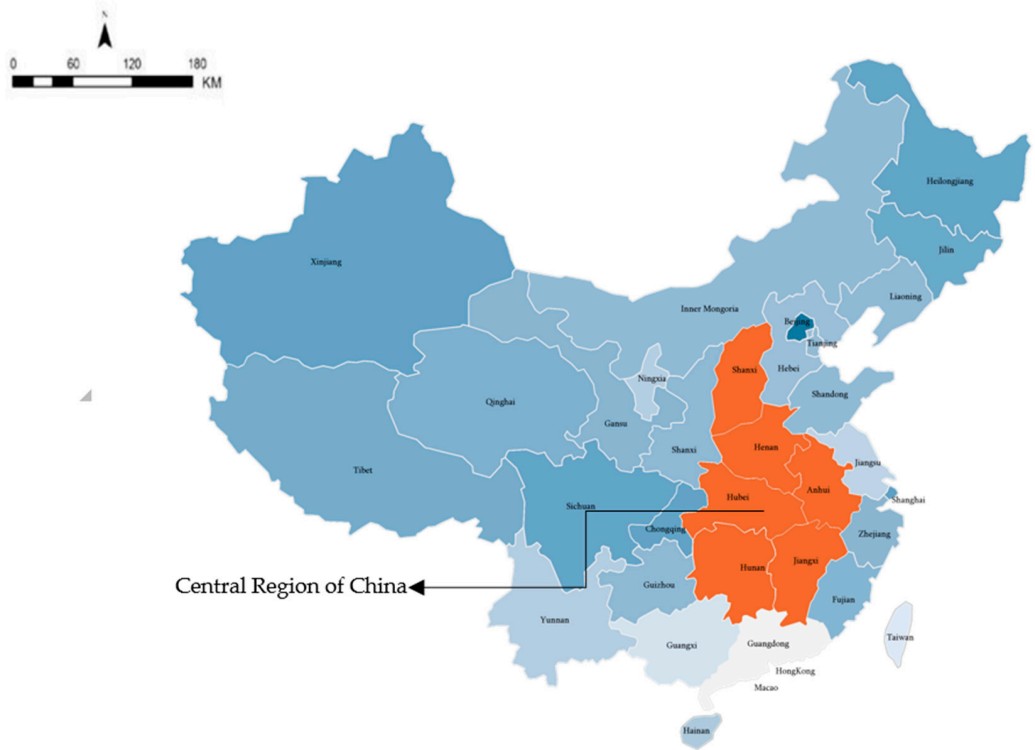

**Figure 1.** Central Region of China (the central regional of China comprise six provinces: Hunan, Hubei, Anhui, Shanxi, Henan, Jiangxi).

*4.2. Hubei Province*

Hubei province is also located in China's central region. It consists of 12 prefecture-level cities and one autonomous prefecture. Known as the 'Land of Fish and Rice', Hubei is rich in agricultural products and mineral resources. Its Gross Democratic Production in 2016 was 3229.8 billion yuan, ranking seventh among the other provinces in China and its growth rate was at 8.1% over that of the previous year, which exceeded the national average. Industrial transformation is also taking place in Hubei Province. According to data published by Hubei Province Bureau of Statistics of China, the GDP proportion of primary sector, secondary sector and tertiary sector in Hubei in 2016 were 11.3%, 44.5% and 44.2%. In 2017, the three sectors accounted for 10.0%, 43.5%, 46.5% of GDP, respectively [38]. However, the proportion of private investment in the total decreased by 1.3%. Hubei also faced some other problems including economic goals not being reached and the transformation of the traditional industries remaining below expectation. Due to the well-established transportation network in Hubei Province and its geographical features, Hubei is well developed in heavy industry, with its harmful environmental effect unfortunately becoming increasingly conspicuous [39,40].

Wuhan is Hubei's capital city as well as a sub-provincial level city (higher than Changsha) and central China's core city [41]. It is a very important transportation hub in China both for passengers and goods, a national famous historical and cultural city and the development base for research and education. Located in the middle reach of the Yangtze River, one of the cradles of Chinese civilization, the cities of Xiangyang, Jingzhou, Jingmen and Ezhou boast being representatives of the original Three Kingdoms culture known from one of China's ancient literary classics. Next to tourism, their manufacturing also accounts for a large share of GDP. While Xiaogan, Jingzhou, Shiyan, Xianning,

Suizhou and Huanggang are rich in natural resources, some of their agricultural products are of national importance. Apart from its primary industry, Suizhou is famous of its motor vehicle production, and was nominated as the 'Chinese Capital of Special Purpose Motor Vehicles' by the Hubei government.

*4.3. Overview of Key Data on Cities in Hunan and Hubei Province*

- Only two cities in Hubei (and none in Hunan) are on pathway 1: Shiyan and Jingmen. The labor input of Shiyan's primary industry is 41.4%, while its GDP only accounts for 12.1% of the three major sectors, indicating that the production efficiency of primary sector is relatively low. The same applies to Jingmen.
- Two cities in Hunan and 11 in Hubei evidence a dominant secondary sector; more than half of these show in fact a dominance of the heavy manufacturing (Table 4: L/H as GDP of Secondary sector in %). The economic output of the heavy manufacturing is higher than that of light manufacturing industry in Hunan province. In Hubei, the number of pathway 2L (five cities) is almost on a par with that of those on pathway 2H (six cities); moreover, the difference in the proportion each type (H and L) makes to GDP is quite small, suggesting that the added value of the heavy and light manufacturing industries are more in balance with each other in Hubei than in Hunan.
- The tertiary sector is more developed than the secondary sector in both provinces, with social services being the leading type of industry. Most public management, security and social organization are non-profit and account for the largest proportion of GDP in social services [42]. In addition, apart from Wuhan, the GDP contribution of scientific research and technical services in Hubei province is low (Appendix D). Qiao has shown that more in general, China's scientific research and technical services lag behind those in developed countries [43].
- Construction and manufacturing are no doubt the pillar industries (Pillar industry refers to the industry or industrial group that occupies an important strategic position in the national economic system, and its industrial scale occupies a large share in the national economy and plays a supporting role) in both provinces (Table 4). The third leading industry in Hubei province is wholesale and retail trade. Only Xianning and Jingzhou have well developed traffic, transportation, storage and post services. While in Hunan Province, the tertiary sector is more diverse, including traffic, transportation, storage and post (Xiangtan and Huaihua), financial mediation (Hengyang, Yueyang, Yiyang, Shaoyang, Yongzhou), and real estate (Zhuzhou), which all contribute significantly to Hunan's economic growth.

Table 4. Key geographic data and economic development pathways for the Hunan Province.

| City | Land Area (km²) | Permanent Population (10,000 Persons) | Three Dominant Industries | GDP/Cap Permanent Population (RMB) | 1/2/3 as GDP (in %) | 1/2/3 as Working Population (in %) | Regional Position | L/H as GDP of Secondary Sector (in %) | P/D/C/S as GDP of Tertiary Sector (in %) | Urban Stage | Expected Pathway | Expected Sub-Pathway |
|---|---|---|---|---|---|---|---|---|---|---|---|---|
| Changsha | 11,816 | 764.5 | Manufacturing (35.8%), Construction (26.2%), Wholesale and Retail Trades (8.7%) | 124,122 | 3.96/48.23/47.8 | 0.08/44.3/55.62 | NAT | 3.8/96.2 | 32.95/21.70/21.32/24.03 | 3/2 | 4/2 | 4P/2H |
| Zhuzhou | 11,307 | 402.2 | Manufacturing (45.5%), Construction (26.8%), Real Estate (5.3%) | 62,081 | 7.92/52.98/39.1 | 0.09/56.1/43.81 | REG | 42.8/57.2 | 30.31/21.56/20.26/27.87 | 2 | 2 | 2H |
| Xiangtan | 5008 | 283.8 | Construction (33.3%), Manufacturing (20.8%), Traffic, Transport, Storage and Post (18.1%) | 65,946 | 8.08/52.29/38.63 | 0.68/42.39/56.93 | REG | 8.9/91.1 | 22.61/18.14/25.65/33.6 | 3/2 | 4/2 | 4S/2H |
| Hengyang | 15,303 | 720.5 | Manufacturing (35.8%), Construction (30.7%), Financial Intermediation (6.7%) | 39,020 | 15.09/41.53/43.38 | 0.14/44.79/55.07 | REG | 32.2/67.8 | 35.02/19.35/15.52/30.12 | 3 | 4 | 4P |
| Shaoyang | 20,830 | 737.5 | Construction (39.3), Manufacturing (24.1%), Financial Intermediation (11.0%) | 20,987 | 21.36/35.53/43.11 | 0.81/37.96/61.23 | REG | 47.5/52.5 | 22.77/18.28/22.34/36.61 | 3 | 4 | 4S |
| Yueyang | 14,858 | 568.1 | Manufacturing (37.7%), Construction (30.0%), Financial Intermediation (7.2%) | 54,832 | 11.15/47.38/41.47 | 1.34/42.56/56.11 | REG | 43.9/56.1 | 27.89/24.11/17.67/30.33 | 3/2 | 4/2 | 4S/2H |
| Changde | 18,190 | 584.5 | Construction (40.9%), Manufacturing (27.0%), Wholesale and Retail Trades (7.0%) | 50,543 | 12.97/42.56/44.46 | 0.12/41.85/58.03 | REG | 63.5/36.5 | 27.21/21.60/23.97/27.22 | 3 | 4 | 4SP |
| Zhangjiajie | 9534 | 153.2 | Construction (31.2%), Manufacturing (14.5%), Hotels and Catering Services (11.5%) | 32,300 | 11.43/21.24/67.33 | 0.57/22.02/77.41 | REG | 54.1/45.9 | 18.90/18.31/29.42/33.37 | 3 | 4 | 4S |
| Yiyang | 12,320 | 439.2 | Construction (35.7%), Manufacturing (34.7%), Financial Intermediation (13.6%) | 33,772 | 18.24/39.76/42.00 | 0.26/41.42/58.32 | REG | 44.4/56.6 | 24.37/20.87/26.05/28.70 | 3 | 4 | 4S |
| Chenzhou | 19,654 | 473.1 | Manufacturing (24.4%), Construction (23.7%), Mining (11.8%) | 46,691 | 9.81/52.06/38.13 | 0.47/37.92/61.61 | REG | 25.7/74.3 | 29.91/25.35/20.41/24.34 | 3/2 | 4/2 | 4P/2H |

**Table 4.** *Cont.*

| City | Land Area (km²) | Permanent Population (10,000 Persons) | Three Dominant Industries | GDP/Cap Permanent Population (RMB) | 1/2/3 as GDP (in %) | 1/2/3 as Working Population (in %) | Regional Position | L/H as GDP of Secondary Sector (in %) | P/D/C/S as GDP of Tertiary Sector (in %) | Urban Stage | Expected Pathway | Expected Sub-Pathway |
|---|---|---|---|---|---|---|---|---|---|---|---|---|
| Yongzhou | 22,260 | 547.9 | Manufacturing (32.9%), Construction (30.9), Financial Intermediation (7.2%) | 28,744 | 20.87/34.83/44.3 | 0.87/33.37/65.76 | REG | 46.8/53.2 | 23.84/19.40/12.54/44.22 | 3 | 4 | 4S |
| Huaihua | 27,758 | 496.0 | Construction (24.3%), Manufacturing (22.0%), Traffic, Transport, Storage and Post (12.5%) | 28,515 | 14.32/38.23/47.46 | 0.52/23.36/76.13 | REG | 30.7/69.3 | 32.75/24.25/18.41/24.59 | 3 | 4 | 4P |
| Loudi | 8117.6 | 365.2 | Construction (40.1%), Manufacturing (30.8%), Mining (7.5%) | 36,058 | 14.72/48.31/35.98 | 0.5/49.95/49.55 | REG | 30.6/69.4 | 21.97/26.11/13.86/38.06 | 2 | 2 | 2H |
| Wuhan | 8569.15 | 1076.2 | Manufacturing (32.9%), Construction (31.2%), Wholesale and Retail Trades (11.3%) | 111,469 | 3.3/43.9/52.8 | 9/38/53 | NAT | 35.4/64.6 | 32.75/10.94/14.87/41.44 | 3 | 4 | 4S |
| Huangshi | 4582.9 | 246.5 | Manufacturing (46.2%), Construction (28.2%), Mining (5.9%) | 53,033 | 8.7/55.3/36 | 20/40.3/39.7 | REG | 17.6/82.4 | 27.71/35.78/12.25/24.26 | 2 | 2 | 2H |
| Shiyan | 23,666 | 340.9 | Manufacturing (45.5%), Wholesale and Retail Trades (21.9%), Construction (12%) | 41,923 | 12.1/47.7/40.2 | 41.4/18.1/40.6 | REG | 66.9/33.1 | 13.29/39.89/7.45/39.37 | 3/2 | 1/4/2 | 1/4DS/2L |
| Yichang | 21,084 | 413.0 | Manufacturing (45.6%), Construction (16.4%), Wholesale and Retail Trades (12.9%) | 89,978 | 10.8/57.2/32 | 25/31.9/43.1 | REG | 40.4/59.6 | 52.87/22.39/5.43/19.31 | 3/2 | 4/2 | 4P/2H |
| Xiangyang | 19,727.68 | 563.9 | Manufacturing (45.7%), Construction (20.6%), Wholesale and Retail Trades (15.8) | 62,134 | 11.7/55.4/32.9 | 1.7/51.7/46.6 | REG | 43.1/56.9 | 7.40/57.59/3.74/31.28 | 2 | 2 | 2 H |
| Ezhou | 1594 | 106.8 | Manufacturing (45.4%), Construction (30.1%), Wholesale and Retail Trades (4.4%) | 74,983 | 12.2/54.5/33.3 | 30.4/31.7/37.9 | REG | 25.8/74.2 | 15.95/14.02/4.67/63.36 | 3/2 | 4/2 | 4S/2H |
| Jingmen | 12,404 | 290.1 | Manufacturing (48.8%), Construction (16.7%), Wholesale and Retail Trades (11.8%) | 52,470 | 14/51.9/34.1 | 37.3/24.9/37.8 | REG | 45.9/55.1 | 11.25/13.52/5.88/69.34 | 3/2 | 4/2/1 | 4S/2H/1 |

**Table 4.** *Cont.*

| City | Land Area (km²) | Permanent Population (10,000 Persons) | Three Dominant Industries | GDP/Cap Permanent Population (RMB) | 1/2/3 as GDP (in %) | 1/2/3 as Working Population (in %) | Regional Position | L/H as GDP of Secondary Sector (in %) | P/D/C/S as GDP of Tertiary Sector (in %) | Urban Stage | Expected Pathway | Expected Sub-Pathway |
|---|---|---|---|---|---|---|---|---|---|---|---|---|
| Xiaogan | 8904.41 | 490.4 | Manufacturing (42.3%), Construction (26.8%), Wholesale and Retail Trades (10.9%) | 32,236 | 17.8/48/34.2 | 1/55.8/43.2 | REG | 44.3/55.7 | 10.61/32.35/13.39/43.66 | 2 | 2 | 2H |
| Jingzhou | 14,067 | 569.7 | Manufacturing (45.6%), Construction (21.2%), Traffic, Transport, Storage and Post (6.4) | 30,305 | 22.2/42.6/35.2 | 28.8/28.7/42.5 | REG | 55.5/44.5 | 29.71/28.53/16.31/25.44 | 3/2 | 4/2 | 4P/2L |
| Huanggang | 17,457.2 | 632.1 | Manufacturing (45.9%), Construction (30.8%), Wholesale and Retail Trades (5%) | 27,373 | 22.9/37.9/39.2 | 3.35/58.42/38.23 | REG | 74.2/25.8 | 13.35/11.89/2.99/71.77 | 2/3 | 2/4 | 2L/4S |
| Xianning | 10,033 | 252.6 | Manufacturing (42.3%), Construction (26.3%), Traffic, Transport, Storage and Post (7%) | 44,027 | 16.6/47.6/35.7 | 27.1/24.3/48.6 | REG | 58.6/41.4 | 12.51/21.16/8.06/58.27 | 3/2 | 4/2 | 4S/2L |
| Suizhou | 9636 | 220.2 | Manufacturing (45%), Construction (28.6%), Wholesale and Retail Trades (8.4%) | 38,801 | 16.5/46.8/26.7 | 0.4/42.2/57.4 | REG | 57.2/42.8 | 8.69/17.68/3.96/69.67 | 3/2 | 4/2 | 4S/2L |

## 5. Analysis

*5.1. City Brand Identities in Hubei and Hunan*

City brand identities are closely related to how cities define themselves in contradistinction to others [44,45]. A city brand identity can be established from self-descriptions of the city in official government documents. The city brand identities of cities in Hunan and Hubei are shown in Table 5. We compared the pathways following geographic position (expected pathways) with brand identities extracted from planning documents, including 12th and 13th Five Year Plans and Urban Master Plans (adopted pathways) and the following observations stand out:

- When it comes to branding, tourism features quite prominently in both provinces. Tourism has a reputation of being ecologically friendly rather than resource-extracting or contaminating, unlike manufacturing, which suggests it is more in line with what ecological modernization requires. This is a first reason for cities to highlight this aspect; another is that much city promotion, marketing and branding was traditionally done for tourism purposes and part of this routine practice has been preserved [46].

- A cultural image is another important aspect in the formation of city brand identities. Nine out of 12 cities in Hubei province and eight out of 14 cities in Hunan province make reference to being a "cultural city" in their planning documents. As an intangible asset, culture also is believed to be characterized by low input of resources and high-value output. Moreover, culture enhances the attraction of high-profile visitors, thus generating economic value [47,48]. Jingzhou, for instance, cherishes Guanyu, a protagonist in the Three Kingdoms.

- Regarding the sub-industries within the tertiary sector, "social service industries" appear to flourish in both provinces in their geographic profiles. However, when it comes to branding in both brand identities and city labels, the mention made of "distribution services" and, to a lesser extent, "consumer services", prevails. A likely reason for this phenomenon is that most of the economic activity involved in social services (except research) counts more as social welfare oriented rather than as GDP-enhancing and is consequently less attractive for branding purposes [42]. Distributive services including wholesale and retail trade, traffic, transport, storage and post, on the other hand, provide obvious comparative advantage in terms of added economic value and constitute therefore an appealing aspect for urban governments to focus on in their branding strategy, especially in central China where having hubs for transport and logistics in place is of key importance [49–51].

- When it comes to the secondary sector, most cities in Hunan and Hubei provinces prefer to define themselves more vaguely as "important industrial base" or "new/new-type industrial base". Only a few cities actually use "manufacturing base" to brand themselves. This is likely the result of the negative ecological impact the secondary sector has or is believed to have. Moreover, cities on pathway 2, such as Huangshi, Xiangyang, Zhuzhou, and Loudi, are more inclined to combine pathway 2 with the seemingly eco-friendlier pathway 4 to convey a cleaner impression of themselves. The same phenomenon was also observed in earlier work regarding the developmental pathways in China's three megacity regions (Jing-Jin-Ji, Yangtze River Delta and Pearl River Delta) and the three Northeastern provinces (Heilongjiang, Jilin and Liaoning) [20,21].

**Table 5.** Developmental pathways and brand identities for cities in Hubei and Hunan.

| City in Hubei | Predicted Pathway | Brand Identity Description (Source) | Adopted Pathway | Corresponding Industries |
|---|---|---|---|---|
| Wuhan | 4S | Wuhan is the capital city of Hubei Province, a national historical and cultural city. It is also an important industrial base, science and education base, and comprehensive transportation hub in China. The goal of economic development is to adjust and optimize the economic structure so as to form an industrial development pattern with high-tech industries as the forerunner and advanced manufacturing and modern service industries as the support. (Urban Master Plan is abbreviated as "UMP"). | 4DC/2H | Distributive + Customer/Heavy Manufacturing |
| Huangshi | 2H | To build Huangshi as a national ecological garden city and transport hub and strategic platform. Huangshi is also a mining and metallurgical culture city and advanced manufacturing base. The goal is to build Huangshi into an "ancient capital of mining and metallurgy" and a "landscape city" (UMP). | 4DC/2H | Distributive + Customer/Heavy Manufacturing |
| Shiyan | 1/4DS/2L | Zhuzhou is an important transportation hub in the Chang-Zhu-Tan City Agglomeration. It is known as a historical and cultural commemorative site for Chinese and overseas Chinese and an eco-garden city. The overall development goal is to build Shiyan into an internationally renowned ecological cultural tourism area, an important national automobile industry base and an ecological liveable city (UMP). | 4DC/2H | Distributive + Customer/Heavy manufacturing |
| Yichang | 4P/2H | Relying on the Yangtze River, Three Gorges dam and giving full play to its resource advantages and build Yichang into a world-famous hydropower tourism city and a regional transportation and circulation center. Yichang will be built as an important manufacturing base in the middle and upper reaches of the Yangtze River and an important financial, cultural, educational, scientific, health, and information service base in the Hubei Province (UMP). | 4DCS/2H | Distributive + Customer + Social/Heavy Manufacturing |
| Xiangyang | 2H | Xiangyang is a national historical and cultural city. It will be built into an important transportation hub and regional logistics center. Economic development strategy: focus on building the automobile industry as the leading industry (UMP). | 4DC/2H | Distributive + Customer/Heavy manufacturing |
| Ezhou | 4S/2H | Ezhou is the core city of Wuhan city circle and the central city of eastern Hubei city cluster, a provincial historical and cultural city, an ecological tourism resort, a green manufacturing base, a regional logistics center and a transportation hub (UMP). | 4DC/2H | Distributive + Customer/Heavy Manufacturing |

**Table 5.** *Cont.*

| City in Hubei | Predicted Pathway | Brand Identity Description (Source) | Adopted Pathway | Corresponding Industries |
|---|---|---|---|---|
| Jingmen | 4S/2H/1 | Jingmen is the central city in Hubei Province, an emerging industrial city featuring petrochemicals and electricity, a provincial historical and cultural city, and a liveable city with good ecological environment (UMP). | 2H/4C/1 | Heavy manufacturing/Consumer service/Primary industry |
| Xiaogan | 2H | Xiaogan is the sub-center city in Wuhan City Circle, the center city both in Hubei and Hunan provinces. The goal is to build Xiaogan into a Chinese filial piety culture city and a liveable leisure city with water garden features (UMP). | 4C | Customer service |
| Jingzhou | 4P/2L | It will build Jingzhou into a well-known tourist destination, one of the important transportation hubs in the middle reaches of the Yangtze River, and a central city and ecologically liveable city in Central and South China. (UMP) | 4DC | Distributive + Customer |
| Huanggang | 2L/4S | Huanggang is regional center city, historical and cultural city, green agricultural production and processing base at the provincial level, a new industrial base in the core area of the Wuhan City Circle. At the same time, it is striving to build itself into an important modern manufacturing base in Wuhan City Circle, a provincial green agricultural production and processing base and the cultural and educational city (UMP). | 4C/2L | Customer, Light manufacturing |
| Xianning | 4S/2L | Xianning is the regional trade and logistics center of Hubei and Hunan, the eco-liveable city of Wuhan City Circle, and the famous tourist city of hot springs in China (UMP). | 4DC | Distributive + Customer |
| Suizhou | 4S/2L | Suizhou is a national historical and cultural city. It is an emerging tourist cities and landscape city. (UMP). To build a well-known important tourist destination, distribution center and the province's important agricultural production and processing, export, new energy, and logistics industry bases. The goal is to strongly promote Suizhou on the construction of innovate cities (13th Five Year Plan is abbreviated as "FYP"). | 4DC/2L | Distributive + Customer/Light manufacturing |
| Changsha | 4P/2H | As the capital city of Hunan province, Changsha is an important central city in the middle reaches of the Yangtze River and national historical and cultural city. It is an important industrial and commercial city and transportation hub in the South-Central region (UMP). The goal is to build Changsha into an advanced manufacturing base with international competitiveness (13th FYP). | 4DC/2H | Distributive + Consumer/Heavy Manufacturing |

**Table 5.** *Cont.*

| City in Hubei | Predicted Pathway | Brand Identity Description (Source) | Adopted Pathway | Corresponding Industries |
|---|---|---|---|---|
| Zhuzhou | 2H | Zhuzhou is an important transportation hub in the south of China, the national old industrial base led by high-tech industries. It is also the important center of commerce and modern logistics in the central and south China region. It is an important industrial city in Hunan province, one of central city in the Chang-Zhu-Tan region and the Yan kingdom historical and cultural commemoratives site (UMP). | 2H/4DC | Heavy manufacturing/Distributive + Consumer |
| Xiangtan | 4S/2H | Xiangtan is the central city in Chang-Zhu-Tan region. It is an important industrial, education and tourism city in Hunan province. The functions of towns and cities at all levels are mainly guided by the functions of manufacturing industry, agricultural service and technical support, trade circulation and tourism service (UMP). | 4SC/2H | Social + Consumer/Heavy Manufacturing |
| Hengyang | 4P | Hengyang is already the transportation hub and the central city in the south of Hunan. The development goal for Hengyang is to be an important industrial city, a famous cultural city, a tourism city and also a liveable city (UMP). Hengyang will strive to build an important electronic information, high-end equipment manufacturing base and an important copper industry base in China (13th FYP). | 4DC/2H | Distributive + Consumer/Heavy Manufacturing |
| Shaoyang | 4S | Shaoyang is a second-class center city and a historical city in Hunan province. It is an important central city and transportation hub in central and southern Hunan, based on industry and trade, and is a political, economic, cultural, and information center in Hunan (UMP). | 4DC | Distributive + Consumer |
| Yueyang | 4S/2H | Yueyang is a national historical and cultural city. It is a scenic tourist city. It is a petrochemical industrial base and a modern logistics center in the central region of China. It is the only port that connects the river and ocean in Hunan province. It is also a liveable city that is lakeside in the middle reaches of the Yangtze River (UMP). | 4DC/2H | Distributive + Consumer/Heavy manufacturing |
| Changde | 4SP | Changde is the central city of Northwestern Hunan. It is a comprehensive transportation hub, and an ecologically liveable city. The functions of Changde City including trade and logistics centers, green food bases, tourism service bases and cultural and educational bases in north western Hunan (UMP). | 4DCS/2L | Distributive + Consumer + Social/Light manufacturing |
| Zhangjiajie | 4S | Based on its abundant tourism resource, Zhangjiajie will be built into a well-established international tourism city with good eco-system, harmonious society and beautiful environment by the end of the planning area. The condition of environment and the eco-system is going to be further enhanced and the tourism industry will be better organized (UMP). | 4C | Consumer |

**Table 5.** *Cont.*

| City in Hubei | Predicted Pathway | Brand Identity Description (Source) | Adopted Pathway | Corresponding Industries |
|---|---|---|---|---|
| Yiyang | 4S | Yiyang is the central city in the south of Dongting Lake Ecological Economic Zone, and the sub-center city of Chang-Zhu-Tan Metropolitan Area. It is also a modern new-type industrial city, and a liveable and water and mountain, eco-tourism city. The main functions of the city are the advanced manufacturing base and energy base in Hunan Province (UMP). | 4C/2H | Consumer/Heavy Manufacturing |
| Chenzhou | 4P/2H | Chenzhou is the southern gateway to Hunan Province. It is a provincial-level historical and cultural city, relying on mineral resources, ecological resources, and locational transportation advantages. It has gradually developed into a provincial-level regional center city of Hunan, Guangdong, and Jiangxi provinces. (Website of Municipal People's Government in Chenzhou). | 4DC | Distributive + Consumer |
| Yongzhou | 4S | Yongzhou is a historical and cultural city in Hunan Province. It is a transportation hub and a central city in the border regions of Hunan, Guangdong, and Guangxi. It is also an ecological city with well-known of water and mountain (UMP). | 4DC | Distributive + Consumer |
| Huaihua | 4P | Huaihua is an important railway transportation hub in China. It is an important central city in the border areas of Hunan, Guangxi, Guizhou, Jiangxi, and Hubei Provinces. (UMP) | 4D | Distributive |
| Loudi | 2H | Loudi is a regional comprehensive transportation hub and new industrialized industrial base in Hunan province. It is the central city in Changsha-Zhuzhou-Xiangtan urban agglomeration. It is also a green and liveable, as well as a tourism city. Loudi will be built into the new energy and raw materials base, characteristic equipment and advanced manufacturing base, culture and ecological tourism and leisure base, regional trade and logistics center, and regional comprehensive transportation hub of Hunan province (UMP). | 4DC/2H | Distributive + Consumer/Heavy manufacturing |

*5.2. City Labels in Hubei and Hunan*

Next to city brand identities, city labels also throw light at city branding practices. City labels are generic phrases cities use to characterize themselves. These labels are often policy-related catchy denominations and are easy to remember [52]. Here as well, we compared expected pathways related to cities' geographic position (Table 4) with adopted pathways as reflected by dominant city labels in the plans (Appendix A). We present the expected and adopted pathways in an overview in Table 6.

- Pathway 4+2 is the most common combination in both provinces. In particular, cities on pathway 2, where the secondary sector is dominant in the industrial profile, seek green washing by making creative combinations with pathway 4. Cities on pathway 4, on the other hand, show high level of congruence between expected and adopted pathways. These findings, too, are consistent with [20,21]. These findings also serve as evidence supporting conclusions drawn in earlier studies, which state that ecological modernization is effective in realizing a relative reduction in the emission of pollutants per capita when tertiary sector industries phase out secondary sector ones; this insight has become common knowledge among policy makers [2]. It is intriguing that Hengyang, Shaoyang, Chengde, and Yiyang, all of which are on pathway 4, also emphasize their secondary sector industries and turn themselves into pathway 4+2 rather than just pathway 4. As a percentage of both working population and GDP, the gap between tertiary and secondary industries in these cities is quite small (see Table 4), which suggests that their secondary sector still plays an important role and they wish not to lose it. Nonetheless, for all municipal governments involved, balancing economic growth and ecological protection in actual industrial regulation and in their branding practices remains a vital challenge.
- Regarding the secondary sector, cities heavy manufacturing is dominant, such as Ezhou, Jingmen and Xiaogan, prefer "light manufacturing" for their branding strategies. Here, industrial reality and branding practice do not correspond with each other. This is especially conspicuous in Ezhou city, where heavy manufacturing accounts for even 74.2% of its GDP. Scholars have pointed out that light sub-industries performed better than heavy sub-industries in terms of ecological efficiency, and generate less natural resource consumption [53–56]. This may explain why cities prefer light manufacturing over heavy manufacturing in their city branding practices. In contradistinction, cities on pathway 2L (light manufacturing) adopt pathways consistent with their actual profile. Such is the case in Shiyan, Huangshi, Jingzhou, Huanggang, Xianning, and Suizhou, where light manufacturing prevails, and escapist attitudes are not needed. Cities wishing to engage in ecological modernization can be expected to face difficulties if their branding shows long-term inconsistency with actual development, as is the case when heavy manufacturing is the backbone of their economy and transformation to light manufacturing is promised, but not delivered. This will negatively affect their credibility in the eyes of their stakeholders such as investors and residents, since urban governments fail to deliver on their brand promises [57].
- Regarding the tertiary sector in both provinces, many cities prefer distributive services over social services in their choice of city labels. Distributive services play a positive role in economic growth, which explains its branding popularity [58,59]. Public management, on the other hand, is non-profit and takes a large share of the workforce, but its contribution to GDP is comparatively low. It should therefore come as no surprise that cities prefer to brand themselves as being on pathway 4D (distributive services) rather than on 4S (social services). Seen from an environmental perspective, however, China's distribution services—especially logistics—generate high and rising levels of carbon emission year after year. However, due to the high economic value of the logistics industry, cities warmly welcome the development of this type of industry, nonetheless. Here, the balance appears to be in favor of economic growth rather than ecological preservation. The only way out from an environmental point of view is to realize that even relative decoupling is improving the energy efficiency of logistics services. Producer services are in fact well-developed in the cities of Hengyang, Huaihua and Chenzhou, although they promote themselves as being

strong in distributive and consumer services. We may surmise that this is due to the fact that in recent years consumer services and distributive services have shown faster economic growth in China than producer services and are therefore more appealing industries to have inside a city's borders [51].

- The trend towards ecological modernization is again confirmed in that among the most frequently mentioned city labels, apart from service city, "advanced manufacturing city" and "innovation city" rank high, reflecting a wish for cities to realize industrial upgrades. In the meantime, cities in Hunan and Hubei also experience industrial transformation from the secondary to tertiary sectors. This is only in line with the precepts of ecological modernization if this involves at least a relative decoupling between rising economic value added and environmental harm done. This is frequently, but not always, the case. Distributive services are embraced more easily than social services in spite of their high levels of energy consumption and carbon emissions and may not represent ecological improvement when compared with certain manufacturing or other secondary sector activities. Finally, as important producers of wheat and rice, cities in Hubei appear barely to value the crucial role of their primary sector. This is fully understandable in terms of economic value added but increases the nation's vulnerability to dependence on food imports.

**Table 6.** The comparison of expected and adopted pathways for Hunan and Hubei Province.

| City | Predicted Pathway | Predicted Corresponding Industry | Adopted Pathway | Adopted Corresponding Industry |
|---|---|---|---|---|
| Zhangjiajie | 4S | Social service | 4S | Social service |
| Yongzhou | 4S | Social service | 4C | Consumer service |
| Wuhan | 4S | Social service | 4D | Distributive service |
| Shaoyang | 4S | Social service | 4D/2L | Distributive service/Light manufacturing |
| Yiyang | 4S | Social service | 4D/2L | Distributive service/Light manufacturing |
| Xiaogan | 2H | Heavy manufacturing | 4D/2L | Distributive service/Light manufacturing |
| Huangshi | 2H | Heavy manufacturing | 4D/2H | Distributive service/Heavy manufacturing |
| Xiangyang | 2H | Heavy manufacturing | 4D/2H | Distributive service/Heavy manufacturing |
| Loudi | 2H | Heavy manufacturing | 4D/2H | Distributive service/Heavy manufacturing |
| Zhuzhou | 2H | Heavy manufacturing | 4D/2H | Distributive service/Heavy manufacturing |
| Huaihua | 4P | Distributive service | 4D | Distributive service |
| Hengyang | 4P | Producer service | 4S/2H | Social service/Heavy manufacturing |
| Changsha | 4P/2H | Producer service, Heavy manufacturing | 4DP/2H | Producer service + Distributive service/Heavy manufacturing |
| Yichang | 4P/2H | Distributive service/Heavy manufacturing | 4D/2H | Distributive service/Heavy manufacturing |
| Chenzhou | 4P/2H | Distributive service/Heavy manufacturing | 4D | Distributive service |
| Jingzhou | 4P/2L | Distributive service/Light manufacturing | 4D/2L | Distributive service/Light manufacturing |
| Xiangtan | 4S/2H | Social service/Heavy manufacturing | 4D | Distributive service |
| Yueyang | 4S/2H | Social service/Heavy manufacturing | 4D/2H | Distributive service/Heavy manufacturing |
| Ezhou | 4S/2H | Social service/Heavy manufacturing | 4D/2L | Distributive service/Light manufacturing |

**Table 6.** *Cont.*

| City | Predicted Pathway | Predicted Corresponding Industry | Adopted Pathway | Adopted Corresponding Industry |
|---|---|---|---|---|
| Jingmen | 4S/2H/1 | Social service/Heavy manufacturing/primary industry | 4D/2L | Distributive service/Light manufacturing |
| Xianning | 4S/2L | Social service/Light manufacturing | 4D/2L | Distributive service/Light manufacturing |
| Suizhou | 4S/2L | Social service/Light manufacturing | 4S/2L | Social service/Light manufacturing |
| Changde | 4SP | Social service + Distributive service | 4D/2L | Distributive service/Heavy manufacturing |
| Shiyan | 1/4DS/2L | Primary industry/Social service/Light manufacturing | 4D/2L/1 | Distributive service/Light manufacturing/Primary industry |
| Huanggang | 2L/4S | Light manufacturing/Social service | 4D/2L | Distributive service/Light manufacturing |

## 6. Conclusions and Discussion

The starting point of this study has been that many modern cities, including those in China, currently face the dual pressure of having to accommodate both industrial restructuring and climate change. This implies that they remain committed to making an attempt at ecological modernization in order to improve their public image and attract investors, high-tech companies, top talents and prosperous visitors. Seen in this light, a shift from the secondary to the tertiary economic sectors appears at first sight to be a two-edged sword: continued, or even rising, economic growth and reduction of environmental harm may go hand in hand. To examine whether industrial transformation is indeed evolving as favorable as that, we have been leaning on an analytical framework in which different developmental pathways for cities can be distinguished based on the international, national or regional status of a city and the dominance of either the primary, secondary or tertiary economic sectors. Following that, we applied it to all cities in the central Chinese provinces of Hubei and Hunan where industrial restructuring of this sort is occurring in full swing. We found that in most cities undergoing ecological modernization, those in which service-oriented industries already prevailed, the urban governments tended to be satisfied with their status and brand themselves in line with their actual industrial structure. Cities where manufacturing industries still prevailed, on the other hand, often fear the consequences of having a dirty image and cloak themselves as either service-oriented cities or as cities that have a combined manufacturing and service-oriented profile. This finding was significant and confirmed the result of earlier studies conducted in other parts of China.

However, this contribution went a step further than previous studies in that both the secondary and tertiary economic sectors were divided up into sub-sectors to develop a clearer picture of the features of this process of ecological modernization; e.g., which types of industries are replacing which ones exactly and how cities brand these transformations. According to the academic literature, the secondary sector consists of manufacturing, construction, mining and the production and distribution of utilities. Whereas construction and utilities are non-distinctive subsectors in the sense that they fulfil key functions in all cities, mining is generally seen as unattractive. None of these subsectors is therefore significant in any city branding strategies. This does not hold for heavy and light manufacturing, which can be both distinctive for city profiles and have a significant impact on their relative attractiveness. The pattern here appears to be that in cities where light manufacturing dominates economic activity, urban governments tend to brand themselves in line with the industrial profile and developmental pathways they find themselves in. However, in the branding practices of cities where heavy manufacturing functions prevail, their importance is downplayed vis-à-vis that of light manufacturing or any service industries they covet. Since the question whether heavy or light manufacturing makes a higher contribution to economic added value in cities cannot be unambiguously answered, we may assume that this phenomenon occurs for the same reason as why service industries outshine manufacturing industries in city branding practices: a green image (with or without a concomitant green reality) is worth gold in the era of ecological modernization. The picture is more complex, however, when it comes to subdividing the tertiary sector. There, a subdivision

was made into social services, distributive services, producer services and consumer services. All of these proved to be of significant economic importance in most Hubei and Hunan cities. While social services had the most extensive workforce and sometimes the highest share in local GDP, in nearly all cases it is the distributive services that tend to be overrepresented in branding. Producer and consumer services occupy middle positions in this spectrum. This outcome cannot be explained by referring to efforts among urban governments to create a green image to the outside world: rather, it is the appeal of higher added economic value generated by trade, transport and logistics that plays a role here. Such growth potential is not expected from non-profit social services (perhaps with the exception of research activities) or relatively low-tech consumer services. The limited attention paid to producer services in Hubei and Hunan begs more questions, but is presumably the result of the fact that central China is seen as having a strong position as a transport and logistics hub within the country, while most sophisticated producer services are provided in the most developed East and not in the center. Since distributive services tend to be high carbon industries, any transition from light industry or other types of services to distribution may well enhance economic growth but is unlikely to reduce environmental harm—neither absolute nor even as relative decoupling. It therefore does not fit the comfortable picture of ecological modernization and provides evidence that urban governments promoting such a shift prefer to follow economic interests rather than environmental ones when forced to choose.

In summary, the emerging picture is one where cities are keen to select key elements in their branding, which are those aspects of their industrial profile that give them a green and more environmentally friendly image, notwithstanding the fact that they may not always be taking effective action truly to realize this industrial greening. This yearning for a green service-oriented image is muted, nevertheless, by the urge to promote industrial activity with high added value. The latter can be seen in the preference for branding one's distributive service functions over the social ones in spite of the fact that they are obviously the more resource-demanding and exhaust-emitting ones. It also occurs in a certain number of other cities, such as Hengyang, Shaoyang, Changde, and Yiyang, where even particular subcategories within the secondary sector proved more popular than the more 'environmentally friendly' tertiary social services.

The main contribution of this study has been its offering of a more nuanced analytical framework for developmental pathways that more accurately reflects the variety in industrial functions. This makes an understanding of both current economic profile and future prospects more precise and the considerations underlying economic branding behavior easier to grasp. It also demonstrates in a more fine-tuned manner how cities trade off types of industries within the secondary and tertiary sectors against each other in their outward appearance. Additionally, it makes clear how green images are pursued, but not at any price: we have seen multiple cases where expectations of higher economic value added prevail over environmental considerations in city branding if the two appear to be in conflict, even to the point of highlighting certain types of manufacturing over more vulnerable categories of services.

That said, there are enough questions left. Delving more deeply into the various categories of social services, producer services or consumer services would help a great deal in finding those among them that do offer more value added than others and/or that cause less environmental harm. Moreover, in this study, only cities located in the provinces of Hubei and Hunan, 25 altogether, have been examined. Even though the data collection process was highly time-consuming, its representativeness for China as a whole, or even the deindustrializing world, is obviously limited. Different regions in China and elsewhere are undergoing different industrial restructuring processes where different types of economic activities result in different developmental sub-pathways. Only further studies can tell what transformations are in process there, how urban governments brand themselves amidst these transformations and how robust the analytical framework established here turns out in pinpointing and explaining their challenges. Finally, as observed by Lu et al. (2017), the role of the national and provincial governments on municipal branding options and choices is quite substantial—especially in

China [60]. This study has not taken the influence of higher tiers of government on local industrial policies into account. Doing that would certainly provide additional insight into the considerations municipal officials have when developing city brand identities and adopting popular city labels. Since this happens increasingly frequently, higher levels of awareness on how city branding is 'done' and what economic and ecological implications this has in the long term will remain highly relevant for both academics and policymakers.

**Author Contributions:** Conceptualization, M.d.J.; methodology, M.d.J. and M.H.; software, M.H.; validation, M.d.J., M.J.; formal analysis, M.H.; investigation, M.d.J.; resources, M.d.J.; data curation, M.H.; writing—original draft preparation, M.H.; writing—review and editing, M.d.J.; visualization, M.J.; supervision, M.J.; project administration, M.J.; funding acquisition, M.J."

**Funding:** This research has been kindly supported by the National Natural Science Foundation of China (NSFC) (Grant No. 71774042; 71532004); the TU-Delft's Initiative for Mobility and Infrastructures (DIMI) and the Erasmus Initiative for the Dynamics of Inclusive prosperity. The Key Program, National Science Foundation of China (Grant No. 71532004) will cover the costs for publishing in open access. The State Key Program of International Science and Technology Cooperation Foundation of China "Key technologies and demonstrations of comprehensive research on urban energy systems and carbon emissions" (No 2017YFE0101700).

**Conflicts of Interest:** The authors declare no conflict of interest.

# Appendix A

**Table A1.** City labels of categorization of secondary sector in Hubei Province.

| City | Most Frequent City Labels in 12th FYP | Most Frequent City Labels in 13th FYP | Most Frequent City Labels in UMP | Overall Dominant City Label (s) | Adopted Pathway |
|---|---|---|---|---|---|
| Wuhan | Service city 78<br>Innovation city 39<br>Tourism city 24<br>Advanced manufacture city 24<br>Liveable city 23 | Service city 78<br>Innovation city 47<br>Liveable 23<br>Tourism city 15 | Service city 52<br>Tourism city 18<br>Innovation city 18 | Service city 208<br>Innovation city 104<br>Tourism city 57<br>Liveable city 46 | 4 |
| Huangshi | Service city 58<br>Advanced manufacture city 29<br>Innovation city 18<br>Liveable city 13<br>Tourism city 11 | Service city 95<br>Eco city 39<br>Innovation 39<br>Advanced manufacture city 35<br>Liveable city 24 | Tourism city 27<br>Service city 27<br>Advanced manufacture city 18<br>Eco city 17 | Service city 180<br>Advanced manufacture city 82<br>Innovation city 57<br>Eco city 56<br>Tourism city 38<br>Liveable city 37 | 4/2 |
| Shiyan | Service city 38<br>Advanced manufacture city 26<br>Eco city 25<br>Innovation city 19<br>Liveable city 18 | Service city 71<br>Tourism city 41<br>Eco city 39<br>Advanced manufacture city 35<br>Innovation city 31 | Service city 69<br>Advanced manufacture city 35<br>Tourism city 19<br>Eco city 15 | Service city 178<br>Advanced manufacture city 96<br>Eco city 79<br>Innovation city 50<br>Tourism city 38 | 4/2/1 |
| Yichang | Service city 37<br>Innovation city 20<br>Tourism city 20<br>Eco city 13<br>Advanced manufacture city 11 | Service city 101<br>Innovation city 51<br>Eco city 29<br>Tourism city 26<br>Liveable city 24 | Service city 54<br>Advanced manufacture city 15<br>Tourism city 14 | Service city 192<br>Innovation city 71<br>Tourism city 70<br>Eco city 42 | 4 |
| Xiangyang | Service city 45<br>Advanced manufacture city 26<br>Innovation city 18<br>Liveable city 13<br>Eco city 12 | —— | Tourism city 67<br>Service city 46<br>Advanced manufacture city 16 | Service city 91<br>Tourism city 67<br>Advanced manufacture city 42 | 4/2 |
| Ezhou | Service city 22<br>Innovation city 17<br>Advanced manufacture city 15<br>Tourism city 13<br>Liveable city 13 | Service city 137<br>Innovation city 33<br>Tourism city 25<br>Liveable city 22<br>Eco city 21<br>Smart city 18 | Tourism city 10<br>Eco city 5<br>Smart city 4<br>Advanced manufacture city 4 | Service city 159<br>Innovation city 50<br>Tourism city 48<br>Liveable city 35<br>Eco city 26<br>Advanced manufacture city 19 | 4/2 |

**Table A1.** *Cont.*

| City | Most Frequent City Labels in 12th FYP | Most Frequent City Labels in 13th FYP | Most Frequent City Labels in UMP | Overall Dominant City Label (s) | Adopted Pathway |
|---|---|---|---|---|---|
| Jingmen | Service city 21<br>Innovation city 16<br>Advanced manufacture city 13<br>Liveable city 13<br>Tourism city 11 | Service city 51<br>Innovation city 43<br>Advanced manufacture city 36 | Service city 23<br>Tourism city 21<br>Advanced manufacture city 19 | Service city 95<br>Advanced manufacture city 68<br>Innovation city 59<br>Tourism city 33 | 4/2 |
| Xiaogan | Service city 38<br>Tourism city 24<br>Advanced manufacture city 22<br>Innovation city 16<br>Liveable city 13 | Service city 71<br>Innovation city 33<br>Liveable city 31<br>Advanced manufacture city 30<br>Tourism city 26<br>Eco city 23 | Service city 108<br>Tourism city 22<br>Eco city 18<br>Advanced manufacture city 14 | Service city 217<br>Tourism city 72<br>Advanced manufacture city 66<br>Innovation city 49<br>Liveable city 44<br>Eco city 41 | 4/2 |
| Jingzhou | Service city 55<br>Tourism city 26<br>Advanced manufacture city 22<br>Innovation city 22<br>Liveable city 14 | Service city 48<br>Innovation city 24<br>Liveable city 16<br>Eco city 15 | Service city 44<br>Tourism city 42<br>Advanced manufacture city 6 | Service city 147<br>Tourism city 68<br>Innovation city 46<br>Liveable city 30<br>Advanced manufacture city 22 | 4/2 |
| Huanggang | Tourism city 32<br>Service city 30<br>Advanced manufacture city 14<br>Innovation city 14 | Service city 43<br>Advanced manufacture city 20<br>Liveable city 18<br>Innovation 18<br>Eco city 17 | Service city 10<br>Liveable city 3<br>Advanced manufacture city 2 | Service city 63<br>Advanced manufacture city 34<br>Tourism city 32<br>Liveable city 18<br>Innovation city 18 | 4/2 |
| Xianning | Eco city 10<br>Service city 1<br>Smart city 1<br>Innovation city 1 | —— | Service city 49<br>Tourism city 19<br>Advanced manufacture city 13 | Service city 50<br>Tourism city 19<br>Advanced manufacture city 13<br>eco city 10 | 4/2 |
| Suizhou | Service city 24<br>Advanced manufacture city 22<br>Eco city 15 | Service city 38<br>Eco city 27<br>Innovation city 24<br>Liveable city 11<br>Tourism city 10 | Service city 34<br>Tourism city 22<br>Advanced manufacture city 16 | Service city 96<br>Advanced manufacture city 38<br>Tourism city 32<br>Innovation city 24 | 4/2 |

**Table A2.** Urban developmental pathways and city labels for Hunan Province.

| City | Most Frequent City Labels in 12th FYP | Most Frequent City Labels in 13th FYP | Most Frequent City Labels in UMP | Overall Dominant City Label (s) | Adopted Pathway |
|---|---|---|---|---|---|
| Changsha | Innovation city 70<br>Service city 58<br>Advanced manufacture city 31<br>Tourism city 16 | Service city 80<br>Innovation city 48<br>Advanced manufacture city 32<br>Tourism city 24 | Tourism city 12<br>Service city 6<br>Advanced manufacture city 6 | Service city 144<br>Innovation city 118<br>Advanced manufacture city 69<br>Tourism city 40 | 4/2 |
| Zhuzhou | Service city 42<br>Innovation city 36<br>Advanced manufacture city 20<br>Eco city15 | Service city 84<br>Innovation city 60<br>Advanced manufacture city 43<br>Tourism city 19 | Service city 32<br>Advanced manufacture city 23<br>Eco city 13 | Service city 198<br>Innovation city 96<br>Advanced manufacture city 86<br>Eco city 28 | 4/2 |
| Xiangtan | Service city 33<br>Tourism city 17<br>Eco city 16<br>Innovation city 16<br>Advanced manufacture city12 | Service city 46<br>Tourism city 29<br>Innovation city 19<br>Eco city 4 | Service city 46<br>Tourism city 28<br>Advanced manufacture city 19<br>Eco city 16<br>Innovation city 11 | Service city 125<br>Tourism city 75<br>Innovation city 46<br>Eco city 36<br>Advanced manufacture city 31 | 4 |
| Hengyang | Service city 48<br>Advanced manufacture city 32<br>Innovation city 28<br>Tourism city 16 | Service city 59<br>Advanced manufacture city 28<br>Innovation city 27<br>Tourism city 21 | Service city 4<br>Advanced manufacture city 1 | Service city 111<br>Advanced manufacture city 61<br>Innovation city 55<br>Tourism city 37<br>Innovation city 34 | 4/2 |
| Shaoyang | Service city 19<br>Advanced manufacture city 15<br>Tourism city 9 | Service city 121<br>Advanced manufacture city 34<br>Tourism city 26<br>Eco city 25 | Service city 20<br>Advanced manufacture city 17<br>Tourism city 12 | Service city 160<br>Advanced manufacture city 66<br>Tourism city 47 | 4/2 |
| Yueyang | Service city 26<br>Liveable city 7<br>Advanced manufacture city 6 | Service city 50<br>Tourism city 20<br>Innovation city 19 | Service city 21<br>Tourism city 21<br>Advanced manufacture city 12 | Service city 97<br>Tourism city 41<br>Innovation city 19<br>Advanced manufacture city 18 | 4/2 |
| Changde | Service city 32<br>Innovation city 21<br>Advanced manufacture city 20<br>Tourism city 10 | Service city 61<br>Advanced manufacture city 28<br>Innovation city 26<br>Smart city 15 | Service city 52<br>Tourism city 22<br>Advanced manufacture city 7 | Service city 145<br>Advanced manufacture city 55<br>Innovation city 47<br>Tourism city 32<br>Smart city 25 | 4/2 |
| Zhangjiajie | —— | Service city 29<br>Tourism city 20<br>Eco city17 | Service city 50<br>Tourism city 24<br>Advanced manufacture city 5 | Service city 79<br>Tourism city 44<br>Eco city 17 | 4 |

**Table A2.** *Cont.*

| City | Most Frequent City Labels in 12th FYP | Most Frequent City Labels in 13th FYP | Most Frequent City Labels in UMP | Overall Dominant City Label (s) | Adopted Pathway |
|---|---|---|---|---|---|
| Yiyang | Service city 16<br>Advanced manufacture city 15<br>Innovation city 12<br>Eco city 9 | Service city 44<br>Advanced manufacture city 22<br>Eco city 14 | Service city 37<br>Advanced manufacture city 15<br>Tourism city 12<br>Eco city 9 | Service city 97<br>Advanced manufacture city 42<br>Eco city 32 | 4/2 |
| Chenzhou | Service city 31<br>Innovation city 21<br>Eco city 13<br>Tourism city 13<br>Liveable city 11 | Service city 79<br>Innovation city 37<br>Tourism city 17<br>Liveable 15 | —— | Service city 110<br>Innovation city 58<br>Tourism city 30<br>Liveable city 26 | 4 |
| Yongzhou | Service city 24<br>Eco city 20<br>Tourism city 20<br>Innovation city 14 | Service city 72<br>Innovation city 26<br>Eco city 22<br>Advanced manufacture city 18<br>Liveable city 17 | Tourism city 47<br>Service city16<br>Advanced manufacture city 11<br>Innovation city 11 | Service city 112<br>Tourism city 67<br>Eco city 42<br>Innovation city 40<br>Advanced manufacture city 29<br>Liveable city 17 | 4 |
| Huaihua | Service city 24<br>Advanced manufacture city 17<br>Liveable city 10<br>Innovation city 9<br>Eco city 8 | Service city 99<br>Innovation city 37<br>Eco city 30<br>Smart city 28<br>Liveable city 26 | Service city 9<br>Eco city 4 | Service city 132<br>Innovation city 44<br>Liveable city 36 | 4 |
| Loudi | Service city 48<br>Innovation city 24<br>Tourism city 14<br>Advanced manufacture city 13<br>Liveable city 13<br>Eco city 11 | Service city 64<br>Innovation city 32<br>Advanced manufacture city 27<br>Liveable city 19 | Service city 38<br>Eco city 15<br>Advanced manufacture city 11 | Service city 150<br>Innovation city 56<br>Advanced manufacture city 51<br>Liveable city 32<br>Eco city 26 | 4/2 |

# Appendix B

**Table A3.** Labels of categorization of secondary sector in Hubei Province.

| Plan | Brand Label as in UMP | | Brand Label as in 13th FYP | | Brand Label as in 12th FYP | | Total | |
|---|---|---|---|---|---|---|---|---|
| City | Light Manufacturing | Heavy Manufacturing | Light Manufacturing | Heavy Manufacturing | Light Manufacturing | Heavy Manufacturing | Dominate INDUSTRY | Sub-Pathway |
| Wuhan | Food industry Textile industry | Petrochemical iron and steel Manufacture Mechanical equipment manufactures Advanced manufacturing | Agrotechnology | Petrochemical advanced manufacturing Intelligent equipment manufacturing | Agrotechnology | Petrochemical Advanced equipment manufacturing High-end equipment manufacturing Advanced manufacturing Mechanical equipment manufacturing Petrochemical Industry | Light manufacturing 9 Heavy manufacturing 51 | 2H |
| Huangshi | Agrotechnology Textile and garment Processing of food from agricultural products | Pharmaceutical Chemicals advanced Manufacturing Metallurgy | Agrotechnology Food and beverage Textile and garment Processing of food from agricultural products | Petrochemical Ferrous metal industry Nonferrous metals industry Advanced equipment manufacturing Intelligent equipment manufacturing High-end equipment manufacturing Pharmaceutical Chemicals Metallurgy Advanced manufacturing | Agrotechnology Food and beverage Textile and garment Processing of food from agricultural products | Petrochemical Metallurgy Advanced manufacturing Pharmaceutical Chemicals | Light manufacturing 58 Heavy manufacturing 74 | 2H |
| Shiyan | Agrotechnology Processing of food from agricultural products | Metallurgy advanced Manufacturing Equipment Manufacturing Industry | Agrotechnology Food and beverage | Equipment Manufacturing Industry Intelligent equipment manufacturing High-end equipment manufacturing Automobile Manufacturing Metallurgy Advanced manufacturing | Agrotechnology | Equipment Manufacturing Industry High-end equipment manufacturing Pharmaceutical Chemicals Metallurgy Advanced manufacturing | Light manufacturing 40 Heavy manufacturing 33 | 2L |
| Yichang | Agrotechnology | Advanced manufacturing | Agrotechnology Food and beverage Processing of food from agricultural products | Advanced manufacturing Equipment Manufacturing Industry High-end equipment manufacturing Auto-parts industry | Agrotechnology Food and beverage Textile industry textile and garment | Equipment Manufacturing Industry Metallurgy Advanced manufacturing Auto-parts industry | Light manufacturing 28 Heavy manufacturing 31 | 2H |

**Table A3.** *Cont.*

| Plan | Brand Label as in UMP | | Brand Label as in 13th FYP | | Brand Label as in 12th FYP | | Total | |
|---|---|---|---|---|---|---|---|---|
| City | Light Manufacturing | Heavy Manufacturing | Light Manufacturing | Heavy Manufacturing | Light Manufacturing | Heavy Manufacturing | Dominate INDUSTRY | Sub-Pathway |
| Xiangyang | Agrotechnology Food industry Processing of food from agricultural products | Metallurgy advanced Manufacturing Pharmaceutical chemicals | Agrotechnology Food industry | Equipment Manufacturing Industry Advanced manufacturing Auto-parts industry | Agrotechnology Food and beverage Food industry Textile and garment | Automobile manufacturing Shipbuilding High-end equipment manufacturing Pharmaceutical chemicals Shipbuilding Metallurgy Advanced manufacturing | Light manufacturing 25 Heavy manufacturing 29 | 2H |
| Ezhou | Agrotechnology Processing of food from agricultural products | N/A | Agrotechnology Processing of food from agricultural products | Advanced manufacturing High-end equipment manufacturing Auto-parts industry | Agrotechnology Textile and garment Processing of food from agricultural products | Advanced manufacturing Auto-parts industry | Light manufacturing 25 Heavy manufacturing 13 | 2L |
| Jingmen | Agrotechnology Processing of food from agricultural products Textile and garment | Equipment Manufacturing Industry Petrochemical Industry Pharmaceutical Chemicals Metallurgy New Manufacturing Industry Advanced manufacturing Petrochemical Auto-parts industry | Agrotechnology Food industry | Petrochemical Equipment Manufacturing Industry Petrochemical Industry Pharmaceutical Chemicals advanced manufacturing Auto-parts industry | Agrotechnology Food industry | Petrochemical Equipment Manufacturing Industry Petrochemical Industry Pharmaceutical Chemicals Advanced manufacturing Auto-parts industry | Light manufacturing 52 Heavy manufacturing 42 | 2L |
| Xiaogan | Agrotechnology Processing of food from agricultural products Food and beverage Textile and garment | Modern Manufacturing Industry Advanced manufacturing High-end equipment manufacturing Auto-parts industry | Agrotechnology Processing of food from agricultural products Food and beverage Food industry Textile and garment | High-end equipment manufacturing advanced Manufacturing Auto-parts industry | Agrotechnology Food industry Textile and garment Processing of food from agricultural products | Automobile manufacturing Metallurgy Advanced manufacturing Auto-parts industry | Light manufacturing 76 Heavy manufacturing 24 | 2L |
| Jingzhou | Agrotechnology Textile and garment Textile industry | Modern Manufacturing Industry Pharmaceutical Chemicals Auto-parts industry | Agrotechnology Processing of food from agricultural products Textile and garment Textile industry | Equipment Manufacturing Industry High-end equipment manufacturing Pharmaceutical Chemicals Advanced manufacturing Auto-parts industry | Agrotechnology Textile and garment Textile industry | Equipment Manufacturing Industry High-end equipment manufacturing Pharmaceutical Chemicals Metallurgy Auto-parts industry | Light manufacturing 55 Heavy manufacturing 41 | 2L |
| Huanggang | Agrotechnology | Modern manufacturing industry | Agrotechnology Food industry Textile and garment industry Textile industry | Advanced equipment manufacturing High-end equipment manufacturing Advanced manufacturing | Agrotechnology Processing of food from agricultural products Food and beverage | Pharmaceutical Chemicals Textile and garment Advanced manufacturing Shipbuilding Auto-parts industry | Light manufacturing 42 Heavy manufacturing 19 | 2L |

**Table A3.** *Cont.*

| Plan | Brand Label as in UMP | | Brand Label as in 13th FYP | | Brand Label as in 12th FYP | | Total | |
|---|---|---|---|---|---|---|---|---|
| City | Light Manufacturing | Heavy Manufacturing | Light Manufacturing | Heavy Manufacturing | Light Manufacturing | Heavy Manufacturing | Dominate INDUSTRY | Sub-Pathway |
| Xianning | Agrotechnology Food and beverage Textile industry textile and garment Processing of food from agricultural products Textile industry | Shipbuilding metallurgy New manufacturing Industry | Textile and garment Agrotechnology Food and beverage Processing of food from agricultural products | Advanced manufacturing Auto-parts industry | Agrotechnology | Metallurgy | Light manufacturing 27 Heavy manufacturing 7 | 2L |
| Suizhou | Textile and garment Processing of food from agricultural products | Pharmaceutical Chemicals Automobile manufacturing Auto-parts industry | Agrotechnology Food industry Textile and garment | Equipment Manufacturing Industry Intelligent equipment Manufacturing Metallurgy advanced Manufacturing Auto-parts industry | Agrotechnology Textile and garment | Pharmaceutical Chemicals Advanced manufacturing Auto-parts industry | Light manufacturing 36 Heavy manufacturing 30 | 2L |

**Table A4.** Labels of categorization of secondary sector in Hunan Province.

| Plan | Brand Label as in UMP | | Brand Label as in 13th FYP | | Brand Label as in 12th FYP | | Total | |
|---|---|---|---|---|---|---|---|---|
| City | Light Manufacturing | Heavy Manufacturing | Light Manufacturing | Heavy Manufacturing | Light Manufacturing | Heavy Manufacturing | Dominant INDUSTRY | Sub-Pathway |
| Chasha | Processing of food from agricultural products | Equipment Manufacturing Industry Advanced manufacturing | Agrotechnology Food industry Tobacco, food | Pharmaceutical Chemicals Advanced manufacturing High-end equipment manufacturing Auto-parts industry | Agrotechnology Food industry Textile and garment Tobacco, food | Automobile manufacturing Auto-parts industry Advanced manufacturing Aerospace Manufacturing Industry | Light manufacturing 18 Heavy manufacturing 25 | 2H |
| Zhuzhou | Processing of food from agricultural products | Metallurgy | Agrotechnology Food industry textile and garment Processing of food from agricultural products Textile industry | Metallurgy advanced Manufacturing Petrochemical Industry Auto-parts industry | Agrotechnology Textile and garment Textile industry | Equipment Manufacturing Industry High-end equipment Manufacturing Metallurgy Advanced manufacturing Auto-parts industry | Light manufacturing 18 Heavy manufacturing 24 | 2H |

**Table A4.** *Cont.*

| Plan | Brand Label as in UMP | | Brand Label as in 13th FYP | | Brand Label as in 12th FYP | | Total | |
|---|---|---|---|---|---|---|---|---|
| City | Light Manufacturing | Heavy Manufacturing | Light Manufacturing | Heavy Manufacturing | Light Manufacturing | Heavy Manufacturing | Dominant INDUSTRY | Sub-Pathway |
| Xiangtan | Agrotechnology Processing of food from agricultural products | Equipment Manufacturing Industry Metallurgy Advanced manufacturing Auto-parts industry | Agrotechnology Textile industry | Intelligent equipment manufacturing Automobile manufacturing Metallurgy Advanced manufacturing Auto-parts industry | Agrotechnology Food industry Textile industry | Equipment Manufacturing Industry Automobile manufacturing Metallurgy Advanced manufacturing Auto-parts industry | Light manufacturing 15 Heavy manufacturing 35 | 2H |
| Hengyang | Metal Processing Industry Processing of food from agricultural products | Equipment Manufacturing Industry Pharmaceutical Manufacturing Industry | Agrotechnology Textile and garment Furniture Manufacturing Industry Textile industry | Intelligent equipment manufacturing High-end equipment manufacturing Metallurgy Advanced manufacturing Auto-parts industry | Agrotechnology Textile and garment Processing of food from agricultural products Textile industry | Metallurgy Metal Processing Industry Advanced manufacturing Equipment Manufacturing Industry High-end equipment manufacturing Auto-parts industry | Light manufacturing 36 Heavy manufacturing 50 | 2H |
| Shaoyao | Food and beverage Textile and garment Textile industry | Processing and manufacturing | Agrotechnology Processing of food from agricultural products Food industry Textile industry Textile and garment Textile industry | Equipment Manufacturing Industry Metallurgy | Agrotechnology Textile industry | Printing Equipment Manufacturing Industry Pharmaceutical Chemicals Metallurgy | Light manufacturing 41 Heavy manufacturing 23 | 2L |
| Yueyang | Processing of food from agricultural products Food industry Textile industry | Equipment Manufacturing Industry Automobile Manufacturing Petrochemical | Processing of food from agricultural products Agrotechnology Textile and garment Textile industry | Equipment Manufacturing Industry Intelligent equipment manufacturing Petrochemical Petrochemical Industry Metallurgy Advanced manufacturing | Agrotechnology Textile industry | Shipbuilding Automobile manufacturing Metallurgy Petrochemical Auto-parts industry | Light manufacturing 44 Heavy manufacturing 50 | 2H |
| Changde | Agrotechnology Processing of food from agricultural products Textile industry | Pharmaceutical Chemicals Automobile manufacturing Equipment Manufacturing Industry | Agricultural and sideline food processing Agrotechnology Textile and garment Textile industry | Equipment manufacturing industry Automobile manufacturing Advanced manufacturing Auto-parts industry | Agrotechnology Food industry Textile and garment Processing of food from agricultural Products Textile industry | Pharmaceutical Chemicals Metallurgy Advanced manufacturing Equipment Manufacturing Industry | Light manufacturing 61 Heavy manufacturing 37 | 2L |
| Zhang jiajie | Agrotechnology Processing of food from agricultural products | Metallurgy | Agrotechnology | Advanced manufacturing | - | - | Light manufacturing 5 Heavy manufacturing 2 | 2L |
| Yiyang | Agrotechnology Food industry Textile industry Processing of food from agricultural Products textile industry | Metallurgy advanced Manufacturing Mechanical equipment manufacturing Shipbuilding Auto-parts industry | Agricultural and sideline food processing Agrotechnology Textile industry Textile and garment Textile industry | Equipment Manufacturing Industry High-end equipment manufacturing Shipbuilding Auto-parts industry | Agrotechnology Food industry Textile industry | Shipbuilding Equipment manufacturing Industry High-end equipment manufacturing Metallurgy Auto-parts industry | Light manufacturing 74 Heavy manufacturing 33 | 2L |

**Table A4.** *Cont.*

| Plan | Brand Label as in UMP | | Brand Label as in 13th FYP | | Brand Label as in 12th FYP | | Total | |
|---|---|---|---|---|---|---|---|---|
| City | Light Manufacturing | Heavy Manufacturing | Light Manufacturing | Heavy Manufacturing | Light Manufacturing | Heavy Manufacturing | Dominant INDUSTRY | Sub-Pathway |
| Yongzhou | Processing of food from agricultural products | Metallurgy Mineral exploitation | Agrotechnology Processing of food from agricultural products | Equipment Manufacturing Industry High-end equipment Manufacturing Auto-parts industry | Agricultural and sideline food Processing Agrotechnology Textile industry | Pharmaceutical Manufacturing Industry New Energy Industry Automobile manufacturing Metallurgy Advanced manufacturing | Light manufacturing 24 Heavy manufacturing 15 | 2L |
| Chenzhou | N/A | N/A | Food industry Agricultural and sideline food processing Agrotechnology | Non-metallic mineral products Pharmaceutical/ Special equipment/ General equipment/ Equipment Manufacturing Industry Manufacturing of computers, communications and other electronic equipment Automobile/ Advanced manufacturing | Agrotechnology Food processing | Equipment Manufacturing Industry Metallurgy Advanced manufacturing | Light manufacturing 34 Heavy manufacturing 20 | 2L |
| Huaihua | Processing of food from agricultural Products Textile industry | Metallurgy | Agrotechnology Food industry Textile industry | Auto-parts industry | Agrotechnology Textile industry | Furniture manufacturing Industry Equipment manufacturing Industry | Light manufacturing 29 Heavy manufacturing 7 | 2L |
| Loudi | Agrotechnology | Equipment Manufacturing Industry Advanced manufacturing Auto-parts industry | Agrotechnology Textile and garment | Equipment Manufacturing Industry High-end equipment Manufacturing Metallurgy Advanced manufacturing Auto-parts industry | Agrotechnology Textile and garment | Equipment manufacturing Industry High-end equipment manufacturing Metallurgy Advanced manufacturing Auto-parts industry | Light manufacturing 30 Heavy manufacturing 39 | 2H |

# Appendix C

**Table A5.** Labels of categorization of the tertiary sector in Hubei Province.

| Plan | Urban Master Plan | | | | 13th Five-Year Plan | | | |
|------|-------------------|---|---|---|---------------------|---|---|---|
| City | Social Service | Producer Service | Distributive Service | Consumer Service | Social Service | Producer Service | Distributive Service | Consumer Service |
| Wuhan | Service Center<br>Modern Service Center | Finance and Trade Zone<br>Financial Center<br>Economic Center | Transportation Hub<br>Public Transit Hub<br>Railway Hub<br>Node Traffic<br>Passenger Transport Center<br>Passenger Terminal/Traffic<br>Traffic Corridor<br>City of Logistics<br>Logistics Base/Center/ Information Center<br>Port Town<br>Shipping Center | Cultural City<br>Recreation and Entertainment<br>Culture and Sports | Service Center<br>Service Town<br>Service City<br>Service Base<br>Pilot Service<br>Service Demonstration City<br>Service Domains<br>Service Centers | Business Center<br>Trade Platform<br>Trade Pilot Zone<br>National Business<br>Logistics Center<br>Financial Asset Transactions<br>Financial Market<br>Financial Center<br>Financial Field<br>Economic Development Zone<br>Economic Pilot<br>Economic Park | Transportation Hub<br>Node/Gateway City<br>National Railway Network Center<br>Transit Metropolis<br>Shipping Center<br>Hub City<br>Railway Center/Hub<br>Passenger Transport Center<br>Passenger Terminal<br>Traffic Center/Corridor<br>Logistics Base/Center/Hub/Park | Cultural City<br>Leisure City<br>Service to Household<br>Culture and Sports |
| Huangshi | Service Center | Business Center | Rail Transport<br>Passenger Transport Center<br>Passenger Terminal<br>Passenger Traffic<br>Transportation Hub<br>Traffic Corridor<br>Public Transit Hub<br>Logistics Center<br>Logistics Park<br>Logistics Industrial Park | Cultural City | Service Center<br>Service City<br>Service Demonstration Zone<br>Pilot Service<br>Service Domains<br>Service Industry Base<br>Service Industry Park | Business Center<br>Trade Pilot Zone<br>Financial Center<br>Economic Development Zone<br>Economic Pilot<br>Economic Demonstration City<br>Economic Corridor | Transportation Hub<br>Node/Port City<br>Traffic Corridor<br>Hub Center<br>Logistics Base/Center/Industrial Park/ Platform/Park<br>Passenger Terminal/Traffic<br>Transport Hub<br>Comprehensive Port | Cultural City<br>Recreation and Entertainment |
| Shiyan | Service center<br>Service city<br>Service base<br>Modern Service Centers | Business center<br>Economic center<br>Economic development zone | Transportation hub<br>Public Transit Hub<br>Railway City/Hub<br>Rail transport<br>Node city<br>Passenger transport center /terminal/traffic/corridor<br>Traffic center<br>Life logistics park<br>Logistics Center/Park<br>Transport hub | Culture service<br>Recreation and entertainment<br>Culture and sports<br>Leisure and shopping | Service center<br>Service city<br>Service Industry Cluster | Business center<br>Trade city<br>Business Logistics Park<br>Financial market<br>Financial Center<br>Economic Development/ Demonstration zone<br>Economic Corridor<br>Economic Industrial Park<br>Economic Park | Transportation hub<br>High Speed Railway<br>Economic Corridor<br>Logistics city<br>Logistics base<br>Logistics center<br>Logistics agglomeration area<br>Logistics Industrial Park<br>Logistics park<br>Logistics Information Platform | Culture and sports |

Table A5. *Cont.*

| Plan | Urban Master Plan | | | | 13th Five-Year Plan | | | |
|---|---|---|---|---|---|---|---|---|
| City | Social Service | Producer Service | Distributive Service | Consumer Service | Social Service | Producer Service | Distributive Service | Consumer Service |
| Yichang | Service center Service base | Business center Business Logistics Park Economic center Economic Industrial Park | Transportation hub Logistics center Logistics hub Logistics transportation Logistics Park Hub center Passenger transport center Passenger terminal Passenger corridor | Recreation and entertainment | Service center Service domains Service Industry Base Service agglomeration area | Business center Business city Trade platform Financial center Economic hub Economic development zone Economic Demonstration Base Economic Park | Transportation hub Hub city Railway hub Node city Traffic city City of logistics Logistics base/center/hub/platform/park Port town | - |
| Xiang Yang | Service Center Service Centers | Business Center Financial Center | Transportation Hub Public Transit Hub Hub City Passenger Transport Center Passenger Terminal Passenger Traffic Logistics Center Logistics Park | Culture City Recreation and Entertainment Leisure Shopping | Service Center Service City | Economic Demonstration Base | Node City Logistics Hub Transportation Hub | Culture City |
| Ezhou | Service Center | - | Transportation Hub | Cultural City Recreation and Entertainment | Service Center Service Town Service City Service Base Service Domains | Business Center Trade City Trade Base Industry and Trade Park Financial Asset Transactions Financial Market Financial Center Financial Platform Economic Development Zone Economic Pilot Zone | Gateway/Hub City Transit Metropolis Railway Hub Port Platform Passenger Terminal Traffic Corridor Logistics City/Base/Center/ Hub/ Platform/ Industrial Park Transportation Hub Center/Hub/Pilot/ Demonstration Zone Shipping Demonstration Zone | Service Center Service Domains Service City |
| Xiaogan | Service Center Service Centers | Business Center Financial Center Economic Center Economic Development Zone Economic Industrial Park | Transportation Hub Node/Gateway City National Comprehensive Transport/Public Transit Hub Railway Center/Hub/Corridor Passenger Transport/Center /Terminal/Traffic/Corridor Traffic Corridor Logistics Base/Center/ Hub/Platform/Park Service Logistics Park Shipping Center/Hub | Culture City Leisure City Recreation and Entertainment Culture and Sports | Service Center Service Town Service Domains | Business Center Trade City Financial Market Economic Center Economic Development Zone Economic Demonstration Zone Economic Pilot Zone Economic Industrial Park | Transportation Hub Node City Hub City Traffic Center Traffic Corridor City of Logistics Logistics Center Logistics Hub/Platform Logistics Industrial Park Logistics Agglomeration Area Logistics Park | Service Center Service Domains Service Base |

**Table A5.** *Cont.*

| Plan | Urban Master Plan | | | | 13th Five-Year Plan | | | |
|---|---|---|---|---|---|---|---|---|
| City | Social Service | Producer Service | Distributive Service | Consumer Service | Social Service | Producer Service | Distributive Service | Consumer Service |
| Huang Gang | Service Center | | Passenger Transport Center | Cultural City Recreation and Entertainment | Service Center Service Town Service City Service Base | Business Center Trading City Financial Center/Platform Economic Development Zone Economic Demonstration Zone Economic Industrial Park Financial and Business Center | Transportation Hub Node/Harbor City City of Logistics Logistics Base/Center/Hub /Demonstration Zone /Industrial Park/Park Passenger Terminal Traffic Corridor Port Logistics Park Shipping Center | Service Center Service Town Service Base |
| Xianning | Service Center Service Centers Service City | Business Center Business City Business Base Business and Trade Agglomeration Zone Economic Development Zone | Transportation Hub Node City Gateway City Public Transport/Transit Hub Passenger Transport Center/ Terminal/Traffic/Corridor City of Logistics Logistics Base/Center/Park Port Town | Recreation and Entertainment Culture and Sports | - | | - | - |
| Suizhou | Service Center Service Town | Economic Development Zone | Transportation Hub Node City Hub Center Rail Transport Passenger Transport Center Passenger Traffic/Corridor Traffic Corridor Logistics Center | Culture City Recreation and Entertainment | Service Center Service City Service Domains | Trade Pilot Zone Economic Development Zone Economic Sphere | Transportation Hub Gateway/Hub City Passenger Transport Center Traffic Corridor Logistics Center/Hub /Demonstration City/Industry Base/Park Railway Logistics Park | Culture City Culture and Sports |
| Jingzhou | Service Center | Business City Financial Center | Transportation Hub Public Transit Hub Node City Passenger Terminal/Traffic Traffic Corridor Logistics Center/Park Logistics Transportation Port Logistics Park Comprehensive Port | Culture City Recreation and Entertainment | Service Center | Business City Financial Center | Transportation Hub Node City Public Transit Hub Traffic Corridor Logistics Center/Park Logistics Transportation Port Logistics Park Passenger Terminal Comprehensive Port | Service to Household Culture and Sports |

**Table A5.** *Cont.*

| Plan | Urban Master Plan | | | | 13th Five-Year Plan | | | |
|---|---|---|---|---|---|---|---|---|
| City | Social Service | Producer Service | Distributive Service | Consumer Service | Social Service | Producer Service | Distributive Service | Consumer Service |
| Jingmen | Service center Service domains | Business center Financial center | Transportation hub Logistics center Logistics hub Logistics Park Passenger transport center Passenger traffic Traffic corridor | Culture city Recreation and entertainment Culture and sports | Service center Service town Service city Service domains Service demonstration area | Business center Trade Pilot Zone Economic development zone Economic Demonstration Zone/ City/ Base/pilot /Industrial Cluster | Transportation hub Node city Logistics base Logistics center Logistics industrial Park Logistics park Shipping center | Recreation and entertainment Culture and sports Hotel catering Leisure shopping |

| | 12th Five-Year Plan | | | | Total | | |
|---|---|---|---|---|---|---|---|
| City | Social Service | Producer Service | Distributive Service | Consumer Service | Dominant Services | | Sub-Pathway |
| Wuhan | Service Center Service Town Service Domains | Business Center Optical Valley Financial Agglomeration Area Economic Demonstration Zone Economic Agglomeration Area Economic Sphere High-Speed Railway Economic Agglomeration Area Financial and Business Center Financial Center Economic Development Zone | Transportation Hub Public Transit Hub Hub City/Base/Center/Port Traffic Corridor Logistics Base/Center Logistics Industrial Park/Park Transportation Center Railway Hub Passenger Transport Center Shipping Center | Cultural City | Social Service 46 Distributive Service 130 Producer Service 40 Consumer Service 26 | | 4D |
| Huangshi | Service Center Service City Service Domains Service Demonstration Area | Business Center Financial Market Financial Pilot Financial Agglomeration Area Economic Demonstration Zone Economic Pilot Economic Development Zone | Transportation Hub Transport Hub Hub City Rail Transport Passenger Terminal City of Logistics Logistics Base/Center Logistics Industrial Park/Park | Cultural City Recreation and Entertainment Culture and Sports | Social Service 43 Distributive Service 97 Producer Service 33 Consumer Service 13 | | 4D |
| Shiyan | Service Center Service City Service Platform | Business Center Trade Center Financial Market Economic Demonstration Zone Economic Pilot Economic Development Zone | Transportation Hub Hub City Logistics Center Logistics Park Transport Hub Shipping Demonstration Zone | Culture and Sports | Social Service 68 Distributive Service 98 Producer Service 42 Consumer Service 26 | | 4D |

**Table A5.** *Cont.*

| Plan | Urban Master Plan | | | | 13th Five-Year Plan | | | |
|---|---|---|---|---|---|---|---|---|
| City | Social Service | Producer Service | Distributive Service | Consumer Service | Social Service | Producer Service | Distributive Service | Consumer Service |
| Yichang | Service Center Service City Service Domains | Business Center Business City Financial Center Economic Pilot | Transportation Hub Logistics Park Hub Center Node City Passenger Transport Center Passenger Terminal City of Logistics Logistics Base/Hub | Recreation and Entertainment Service to Household | Social Service 43 Distributive Service 84 Producer Service 44 Consumer Service 10 | | | 4D |
| Xiang Yang | Service Center | Business Center Business and Trade Agglomeration Zone Financial Center Economic Development Zone Economic Pilot/Corridor/City Economic Park/Industrial Park | Transportation Hub/Center Node City City of Logistics Logistics Base/Center/Park Logistics Information Hub Railway Hub | Culture City Recreation and Entertainment Culture and Sports | Social Service 29 Distributive Service 61 Producer Service 16 Consumer Service 31 | | | 4D |
| Ezhou | Cultural city Recreation and entertainment Culture and sports | Business and Trade Agglomeration Zone Industry and trade new town Economic development zone Economic Demonstration Zone Economic pilot | Transportation hub Hub center Logistics base Logistics center Port Industrial Park | Cultural city Culture and sports | Social service 30 Distributive service 66 Producer service 31 Consumer service 26 | | | 4D |
| Xiaogan | Culture City Recreation and Entertainment Service to Household Culture and Sports | Business Center Financial Market Financial Pilot Economic Center Economic Development Zone Economic Demonstration Zone Economic Agglomeration Area Economic Park | Transportation Hub Gateway City Node/ Railway City Hub City/Center Passenger Traffic Logistics Base/Center/Hub/Park Transport Hub/Platform Shipping Center | Culture City Recreation and Entertainment Service to Household Culture and Sports | Social Service 44 Distributive Service 151 Producer Service 38 Consumer Service 39 | | | 4D |
| Huang Gang | Cultural City | Trade City Financial Center Economic Pilot Economic Pilot Zone Economic Corridor Economic Development Zone | Transportation Hub/Center Node City Rail Transport Logistics City/Base/Center Logistics Industrial Park/Park Port Logistics Park Business Logistics Park | Culture and Sports Recreation and Entertainment | Social Service 19 Distributive Service 62 Producer Service 34 Consumer Service 6 | | | 4D |

**Table A5.** *Cont.*

| Plan | Urban Master Plan | | | | 13th Five-Year Plan | | | |
|---|---|---|---|---|---|---|---|---|
| City | Social Service | Producer Service | Distributive Service | Consumer Service | Social Service | Producer Service | Distributive Service | Consumer Service |
| Xianning | - | | - | - | Social Service 8<br>Distributive Service 29<br>Producer Service 13<br>Consumer Service 4 | | | 4D |
| Suizhou | Service Center<br>Service Domains | Financial Market<br>Economic Development Zone<br>Economic Center | Node City<br>Hub City<br>Passenger Transport Center<br>Logistics Base/Center/Platform | Culture City<br>Culture and Sports | Social Service 40<br>Distributive Service 36<br>Producer Service 25<br>Consumer Service 23 | | | 4S |
| Jingzhou | Service Center<br>Service Domains | Business and Trade Agglomeration Zone<br>Trade Center<br>Financial Market<br>Financial Center<br>Financial Agglomeration Area<br>Economic Development Zone<br>Economic Center | Transportation Hub<br>Node/Hub City<br>Passenger Terminal<br>City of Logistics<br>Logistics Base/Center/Hub/Park<br>Logistics Demonstration City<br>Port Logistics Park<br>Logistics Information Platform<br>Transport Hub | Culture City<br>Recreation and Entertainment | Social Service 22<br>Distributive Service 97<br>Producer Service 39<br>Consumer Service 31 | | | 4D |
| Jingmen | Service Center/City<br>Service Demonstration Area<br>Service Sector/Domains | Trade City<br>Economic Development Zone 1<br>Economic Platform<br>Economic Pilot<br>Economic Park | Node City<br>Logistics Base<br>Logistics Center<br>Logistics Park<br>Passenger Transport Center | Culture and Sports | Social Service 29<br>Distributive Service 40<br>Producer Service 34<br>Consumer Service 36 | | | 4D |

**Table A6.** City labels of categorization of tertiary sector in Hunan Province.

| Plan | Urban Master Plan | | | | 13th Five-Year Plan | | | |
|---|---|---|---|---|---|---|---|---|
| City | Social Service | Producer Service | Distributive Service | Customer Service | Social Service | Producer Service | Distributive Service | Customer Service |
| Changsha | N/A | Economic Development Zone | Transportation Hub Passenger Traffic Passenger Terminal Transport Hub | Culture City Recreation and Entertainment | Service Center Service Domains Modern Service Center Service City | Trade City Business City Economic Development Zone Economic Demonstration Zone Economic Corridor | Transportation Hub Node City Passenger Terminal Hub City Hub Center Railway Hub Logistics Base Transport Hub | Culture City Home Service Recreation and Entertainment City of Entertainment Culture and Sports Shopping Capital |
| Zhuzhou | Service Center Service Town | Business Center Trade Center | Transportation Hub Traffic Corridor Public Transit Hub Passenger Traffic Transport Hub Logistics Base | Recreation and Entertainment | Service Center Service Domains Service City | Economic Development Zone Economic Pilot Economic Corridor Financial Market Financial Agglomeration Area | Transportation Hub Passenger Terminal Logistics Base Business Logistics Park | Culture City Service to Household Culture and Sports |
| Xiangtan | Service Center Service Town Service Base | | Transportation Hub Traffic Corridor Passenger Traffic Passenger Terminal | Culture City Recreation and Entertainment Culture and Sports | Service Center Service Domains Service City Service Town Service Base | Economic Center Economic Corridor Financial Market Financial Pilot | Passenger Terminal Transportation Hub Passenger Transport Center | Culture City Culture and Sports |
| Hengyang | | | Transportation Hub Logistics Center Logistics Network Center | - | Service Center Service Domains Service City Service Town Service Base | Economic Development Zone Economic Corridor Financial Market Financial Agglomeration Area | Transportation Hub Logistics Base Service Outsourcing | Culture City Recreation and Entertainment |
| Shaoyang | Service Center | Trade Center Economic Development Zone | Public Transit Hub Transportation Hub Node City Passenger Traffic Passenger Terminal Rail Transport | Culture City Recreation and Entertainment | Service Center Service Domains Service Base | Trade City Business City Economic Center Economic Development Zone Financial Center Financial Market Financial Agglomeration Area | Transportation Hub Public Transit Hub Passenger Terminal Node City Rail Transport Logistics Base Business Logistics Park Passenger Terminal Node City Railway Hub | Culture City Recreation and Entertainment Business and Catering Culture and Sports |
| Yueyang | Service City | Economic Center Economic Development Zone | Transportation Hub Public Transit Hub Passenger Traffic Hub City Transport Hub Gateway City Logistics Base | Culture City Recreation and Entertainment Culture and Sports | Service Center Service Domains Service City | Business City | Transportation Hub Passenger Terminal Hub City Node City Logistics Base | Culture City Recreation and Entertainment Service to Household |

**Table A6.** *Cont.*

| Plan | Urban Master Plan | | | | 13th Five-Year Plan | | | |
|---|---|---|---|---|---|---|---|---|
| City | Social Service | Producer Service | Distributive Service | Customer Service | Social Service | Producer Service | Distributive Service | Customer Service |
| Changde | Service Center Service Base | Trade City Trade Center Economic Development Zone Economic Corridor | Port Hub Public Transit Hub Transportation Hub Traffic Corridor Passenger Traffic Passenger Terminal Hub City Railway City/Hub | Culture City Recreation and Entertainment Culture and Sports | Service Center Service Domains Service City | Business Center Business City Economic Demonstration Zone Financial Center | Transportation Hub Node City Passenger Traffic Passenger Terminal Hub City Rail Transport | Culture City Recreation and Entertainment Culture and Sports |
| Zhangjiajie | Service Center Modern Service Center Service Base | N/A | Transportation Hub Public Transit Hub Passenger Traffic Passenger Terminal Transport Hub | Recreation and Entertainment Culture and Sorts | Service Center Business Center | Economic Park Economic Development Zone Economic Corridor Financial Center | Transportation Hub Traffic Corridor Hub Center Hub City Logistics Base Business Logistics Park | Business and Catering Culture and Sports |
| Yiyang | Service Center Service City | Business Center Trade Center Economic Center Economic Pilot | Public Transit Hub Transportation Hub Traffic Corridor Node City Passenger Traffic Passenger Terminal Hub Center Railway Hub Logistics Base | Recreation and Entertainment Culture City | Service Center Service Domains Service City Service Town Service Base | Economic Corridor | Transportation Hub Hub City Port Logistics Park | Culture City |
| Chenzhou | N/A | N/A | N/A | N/A | Service Center Service Domains Service City Service Town | Economic Center Economic Development Zone Economic Pilot Financial Market | Public Transit Hub Transportation Hub Traffic Corridor Node City Hub City Logistics Base Transport Hub | Culture City Leisure City Recreation and Entertainment Culture and Sports |
| Yongzhou | Service Base | Business Center Business City | Transportation Hub Passenger Terminal Hub Center | Culture City Recreation and Entertainment | Service Center Service Domains Service City | Economic Development Zone Economic Demonstration Zone Economic Pilot Economic Corridor Financial Market Business City | Transportation Hub Node City Passenger Terminal Hub City Railway Hub | Culture and Sports |
| Huaihua | Service Center | Economic Center Economic Development Zone Business Center Business City | Transportation Hub Passenger Traffic Rail Transport Railway Hub | Recreation and Entertainment | Service Center Service Domains Service City | Trade City Economic Center Economic Sphere Financial Center | Transportation Hub Node City Business Logistics Par Hub City Hub Center Rail Transport Railway Hub Logistics Base | Recreation and Entertainment Culture and Sports |

**Table A6.** *Cont.*

| Plan | Urban Master Plan | | | | 13th Five-Year Plan | | | |
|---|---|---|---|---|---|---|---|---|
| City | Social Service | Producer Service | Distributive Service | Customer Service | Social Service | Producer Service | Distributive Service | Customer Service |
| Loudi | Service Center | Business Center Economic Park Economic Development Zone Economic Pilot Economic Corridor | Transportation Hub Traffic Corridor Node City Passenger Traffic Passenger Terminal Passenger Transport Center Hub City Railway Hub Logistics Base Transport Hub | - | Service Center Service Domains Service City | Business Center Economic Development Zone Economic Demonstration Zone Economic Corridor | Transportation Hub Traffic Corridor Node City Business Logistics Park Railway Hub Logistics Base | Recreation and Entertainment Culture and Sports Whole and Retail Trade |

| Plan | 12th Five-Year Plan | | | | Total | | Sub-Pathway |
|---|---|---|---|---|---|---|---|
| City | Social Service | Producer Service | Distributive Service | Customer Service | Dominant Service | | |
| Changsha | Service Center Service City Service Base | Business Center Trade City Economic Development Zone Economic Demonstration Zone Economic Pilot/Corridor/Sphere Financial Center | Transportation Hub Public Transit Hub | Culture City Recreation and Entertainment Business and Catering Service to Households Culture and Sports | Social Service 18 Producer Service 32 Distributive Service 32 Customer Service 31 | | 4DP |
| Zhuzhou | Service Center | Economic Pilot Economic Sphere | Transportation Hub Service Outsourcing | Culture City Recreation and Entertainment Leisure and Shopping | Social Service 30 Producer Service 27 Distributive Service 34 Customer Service 17 | | 4D |
| Xiangtan | Service Center Service Domains Service City Service Base | Economic Pilot Zone | Transportation Hub Passenger Transport Center Logistics Base Business Logistics Park Passenger Transport Center Logistics Demonstration City Passenger Transport Center | Culture City Recreation and Entertainment Culture and Sports Leisure Shopping | Social Service 38 Producer Service 8 Distributive Service 48 Customer Service 35 | | 4D |
| Hengyang | Service Center Service Domains Service City | Economic Pilot/Corridor Business City Economic Center/Park Economic Development Zone Economic Demonstration Zone | Transportation Hub Traffic Corridor Public Transit Hub Hub City Node City | Culture City Recreation and Entertainment Service to Households Culture and Sports | Social Service 29 Producer Service 27 Distributive Service 15 Customer Service 18 | | 4S |

**Table A6.** *Cont.*

| Plan | Urban Master Plan | | | | 13th Five-Year Plan | | | |
|---|---|---|---|---|---|---|---|---|
| City | Social Service | Producer Service | Distributive Service | Customer Service | Social Service | Producer Service | Distributive Service | Customer Service |
| Shaoyang | Service Center | Economic Development Zone<br>Economic Pilot Zone | Transportation Hub<br>Hub City<br>Hub City | Recreation and Entertainment | Social Service 20<br>Producer Service 24<br>Distributive Service 54<br>Customer Service 22 | | | 4D |
| Yueyang | Service Center<br>Service Domains<br>Service City | Trade City<br>Business City | Transport Hub<br>Logistics Base<br>Service Outsourcing | Recreation and Entertainment<br>Culture and Sports | Social Service 13<br>Producer Service 7<br>Distributive Service 31<br>Customer Service 23 | | | 4D |
| Changde | Service Center<br>Service Domains<br>Service City | Economic City/Pilot<br>Economic Development Zone<br>Economic Demonstration Zone<br>Financial Center<br>Financial Market | Transport Hub<br>Rail Transport<br>Railway Hub<br>Transportation Hub<br>Passenger Traffic | Culture City<br>Home Service | Social Service 36<br>Producer Service 28<br>Distributive Service 57<br>Customer Service 21 | | | 4D |
| Zhangjiajie | N/A | N/A | N/A | N/A | Social Service 40<br>Producer Service 8<br>Distributive Service 25<br>Customer Service 17 | | | 4S |
| Yiyang | Service Center<br>Service Domains | Economic Development Zone<br>Economic Pilot<br>Financial Market | Transportation Hub | Culture and Sports | Social Service 24<br>Producer Service 9<br>Distributive Service 28<br>Customer Service 8 | | | 4D |
| Chenzhou | Service Center<br>Service Domains<br>Service City<br>Service Base | Economic Center<br>Economic Development Zone<br>Economic Pilot<br>Economic Sphere | Transportation Hub<br>Hub City<br>Hub City<br>Rail Transport | Culture City<br>Leisure City<br>Culture and Sports | Social Service 15<br>Producer Service 14<br>Distributive Service 20<br>Customer Service 11 | | | 4D |
| Yongzhou | Service Center<br>Service Domains<br>Service City | Economic Development Zone<br>Economic Pilot | - | Culture City<br>Recreation and Entertainment | Social Service 20<br>Producer Service 16<br>Distributive Service 26<br>Customer Service 51 | | | 4C |
| Huaihua | Service Center<br>Service Domains<br>Service City | Financial Center | Transportation Hub<br>Passenger Terminal<br>Rail Transport<br>Business Logistics Park<br>Railway Hub | Recreation and Entertainment | Social Service 15<br>Producer Service 13<br>Distributive Service 61<br>Customer Service 7 | | | 4D |
| Loudi | Service Center<br>Service City<br>Modern Service Base | Business Center<br>Trade City<br>Business City<br>Economic Center<br>Economic Park<br>Economic Development Zone<br>Economic Pilot | Passenger Traffic<br>Hub City<br>Node City<br>Logistics Base<br>Node City<br>Transportation Hub | Culture and Sports | Social Service 10<br>Producer Service 48<br>Distributive Service 58<br>Customer Service 5 | | | 4D |

## Appendix D

Gross Domestic Product by Sector (2016).

(Unit: 100million yuan)

| Cities and Prefecture | Gross Domestic Product (100 million Yuan) | Agriculture, Forestry, Animal Husbandry and Fishery | Industry | Construction | Wholesale and retail Trade | Traffic, Transport, Storage and Post | Accommodation and Restaurants | Finance | Information Transfer, Software and Information Technology Service | Management of water Conservancy, Environment and Public Establishment | Tenancy and Business Services | Scientific Research, Technical Service | Real Estate | Resident Service and Other Services | Education | Health and Social Work | Culture, Sports and Entertainment | Public Management and Social Organization |
|---|---|---|---|---|---|---|---|---|---|---|---|---|---|---|---|---|---|---|
| Changsha | 9356.91 | 378.43 | 3727.24 | 790.95 | 699.31 | 268.65 | 242.10 | 568.43 | 245.66 | 41.82 | 326.49 | 242.96 | 329.44 | 255.37 | 277.53 | 107.93 | 453.68 | 400.94 |
| Zhuzhou | 2488.45 | 201.27 | 1152.19 | 167.67 | 123.84 | 84.69 | 41.55 | 73.45 | 56.99 | 7.84 | 63.26 | 17.07 | 99.51 | 90.59 | 68.70 | 60.87 | 63.80 | 115.15 |
| Xiangtan | 1866.79 | 156.15 | 889.40 | 92.93 | 85.96 | 46.16 | 34.01 | 57.64 | 31.14 | 15.63 | 37.27 | 20.38 | 38.61 | 88.87 | 74.15 | 36.46 | 63.92 | 98.10 |
| Hengyang | 2868.57 | 439.28 | 986.93 | 165.23 | 157.72 | 115.06 | 68.56 | 81.56 | 152.22 | 22.01 | 57.26 | 25.88 | 60.42 | 138.38 | 116.40 | 86.94 | 32.49 | 162.03 |
| Shaoyang | 1530.26 | 330.46 | 457.31 | 87.35 | 80.55 | 39.23 | 31.79 | 55.66 | 18.95 | 7.19 | 11.31 | 7.21 | 63.25 | 97.30 | 69.67 | 55.60 | 17.27 | 100.18 |
| Yueyang | 3100.87 | 350.72 | 1321.68 | 149.56 | 202.22 | 106.14 | 63.34 | 58.98 | 77.84 | 18.96 | 131.04 | 33.96 | 88.88 | 119.02 | 91.84 | 92.57 | 43.59 | 150.53 |
| Changde | 2953.82 | 396.34 | 1128.22 | 129.84 | 154.59 | 126.06 | 52.81 | 77.54 | 76.72 | 24.55 | 128.85 | 20.35 | 70.46 | 160.15 | 82.90 | 77.95 | 98.53 | 147.95 |
| Zhangjiajie | 493.10 | 58.39 | 85.81 | 19.07 | 24.94 | 35.48 | 22.39 | 20.46 | 12.32 | 7.43 | 16.23 | 2.23 | 13.25 | 58.57 | 17.14 | 16.63 | 16.09 | 66.68 |
| Yiyang | 1493.18 | 275.38 | 533.63 | 61.64 | 68.18 | 61.77 | 40.28 | 47.10 | 42.02 | 6.53 | 24.69 | 10.40 | 37.91 | 80.97 | 46.25 | 22.88 | 40.91 | 92.63 |
| Chenzhou | 2204.13 | 219.31 | 1057.13 | 91.34 | 137.63 | 74.36 | 49.46 | 60.42 | 76.31 | 5.90 | 63.80 | 15.19 | 49.59 | 88.59 | 34.80 | 46.80 | 32.67 | 100.85 |
| Yongzhou | 1571.33 | 331.61 | 463.82 | 87.57 | 63.39 | 70.13 | 19.15 | 56.31 | 36.21 | 5.94 | 21.87 | 7.16 | 49.74 | 53.33 | 67.94 | 41.23 | 13.85 | 182.10 |
| Huaihua | 1388.23 | 201.06 | 461.88 | 65.69 | 94.33 | 65.63 | 36.11 | 53.46 | 25.80 | 4.82 | 39.23 | 8.23 | 97.52 | 60.51 | 57.74 | 26.21 | 24.83 | 62.17 |
| Loudi | 1398.17 | 207.23 | 592.75 | 75.80 | 69.13 | 67.27 | 22.62 | 33.03 | 29.88 | 8.00 | 19.05 | 6.03 | 32.84 | 38.16 | 61.14 | 27.63 | 11.61 | 96.03 |

**Figure A1.** Resources: Hunan Statistical Yearbook 2017.

## Appendix E

**Table A7.** Labels and corresponding key words of main pathways.

| Main Pathway | Main City Labels | Subordinate City Labels |
| --- | --- | --- |
| Pathway 1 | Modern agriculture city | Agriculture center/Green food base |
| | Eco city | Eco city/green city/forest city/garden city/green model city/environmental protection model city/water-saving cities/water and mountain city |
| | Liveable city | Liveable city/city with good urban living environment |
| Pathway 2/3 | Smart city | Smart city/intelligent city/information city/digital city |
| | Low carbon city | Low carbon city/recycling economy advanced city/public transport city |
| | Advanced manufacture center/base | Advanced manufacture center/base/high tech base city/electronic information industrial base/equipment manufacturing base/emerging industrial base/headquarter base/clean energy base |
| Pathway 4/5 | Tourism city | Tourism city/history city/culture city/coastal city |
| | Innovation city | Innovation city/knowledge city/city for start-ups /learning city/talent/education city |
| | Service city | Service center for industry/trade center/financial center/transport hub/logistics base/transport base/e-commerce pilot cities/service outsourcing demonstration city/port transport city/shipping center/exhibition center |

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
