# Peer review of "City Branding and Industrial Transformation from Manufacturing to Services: Which Pathways do Cities in Central China Follow?"

_sustainability, doi:10.3390/su11215992_

Round 1
Reviewer 1 Report
Dear Authors,
Thank you very much for this contribution. The research is very interesting and sheds light on the relation between city marketing and the existing economic sectors. In my opinion, the article needs improvement to be considered for publication. Foremost, it is necessary to check the text in grammar and style. For example, each page has the same footnote…
First of all, I would encourage the authors to improve their structure. It would benefit if there is a clear problem statement, a RQs, a literature review and an operationalization, then adapted to the empirics, ending with a reflection on the theory. For example, why is table 1 placed within the introduction? This seems already part of the literature review. Restructuring will help to improve the focus of the paper.
Introduction:
It was noticeable that there no references in the first paragraph, although strong statements are used. In the first sentence, ‘cities aim to update their structures’. What structures? Institutional, economic, spatial, political? (continuing) ‘This is especially true in areas…’ If this is true, I think a reference is necessary, otherwise, why? Conceptually (and even ontologically) there is a gap within the paper. Although the research focuses on city branding, and thus on an administrative bounded unit, it also touch upon ‘the reality’, in this case the urban economy. As you know, this economy of course is not bounded. If a city wants to modernize ecologically, by for example lowering its energy use or heavy industry, this doesn’t imply that these functions disappear and are not anymore necessary for the urban economy taken into consideration. This to say, could the authors also take a more critical point of view on these debates and talk about the consequences of externalizing these functions. Especially in China, this is a huge debate whereby the ‘level 1 cities’ indeed can become idealized places, but at the cost of lower level cities, which get all their undesired functions, with all consequences. This is so to say the ‘dark side of city branding’. A sort of zero sum game. A win-win within a city, yes, a major win-lose considering the network of cities. Is the reference style correct? Sometimes the year is added, sometimes not. Check this please. (line 85-87, 153, 158) If table 1 is derived from an author, please mention this in the caption. Line 103-105: difficult to understand. Is there a comma missing?A revised conceptual framework
Although this should be your literature review, suddenly ‘empirical data’ is mentioned (143). What data? How is this used, or how is this processed. Suddenly Table 2 appears with some important results, but it is not clear where this comes from. Or it is derived from other studies, but then this should be explained clearly. Line 158-160 difficult to read. Who made this modification, the authors or the Chinese NBS? If this is your methodology, this should be clearly put forward as that – again structure!Methodology
This reads like a second paper because it now explains how the research will be conducted. How then did you make up the first 2/3 tables? Is this your research or not? Why are the lettres in your pathways sometimes small capitals? 2B or 2b, that seems to change in the text.Results:
Are good, could maybe need a bit more focus.Conclusion and discussion:
433: ‘Non od’? 452: from not for? Refering to earlier comments, based on 442-443 “a green image Is worth gold in the era of ecological modernization”, could you apply a critical perspective on this? Regarding not all cities (in China) are allowed to ‘choose’ their image, but are forced upon taking the ‘bad stuff’ from the higher ranked cities.Author Response
Dear Reviewer,
Thank you very much for your comments!
We have listened to your opinions carefully and made corresponding modifications to the manuscript. We also wrote an 'Author's reply' for your references, please see the attachment.
Kind Regards,
Meiling Han

Reviewer 2 Report
Dear authors,
The article deals with an interesting topic but fall short to delivers what it promises.
The first major issue concerns the topic of ecological modernization. It is absolutely central in the introduction but then, as a concept, is totally left aside in the empirical study. It is then resumed in the conclusions but without any depth and without any kind of analysis in the empirical section.
In general, the article presents important and valuable data and I can imagine that it required a significant amount of work to collect it. However, the article does little more than the presentation of collected data.
In all tables, I have to raise some concern about how the correspondence of cities with the pathways was made. How was the correspondence between the dominant industry and the respective city done? What is the cutting line that determines that a city falls in a certain pathway and not in another? (For instance, the first two cities (Changsha and Zhuzhou) share the same dominant industry but with different predicted pathways. Regarding this subject, the tables 4 – 6 are not reader friendly. The authors are leaving to the reader the onus of reading, extracting and interpreting the information in these tables that all together occupy 10 pages. My recommendation is that some synthesized tables or graphs are introduced in the text to replace those tables (that can be placed as attachments)
There is a lack of critical analysis of presented information. As it is, this article is more a source of information than a scientific article. What is mentioned in line 429 “this contribution went a step further in subdividing both the secondary and tertiary economic sectors into sub.sectors”, is not sufficient and is more appropriated to a working paper.
I recommend the authors to make an effort to make the discussion about what does the collected data means; what are the implications for the ecological modernization?; What implications in terms of knowing if the analysed data means a path towards the “absolute decoupling” or “relative decoupling”.
Hope you may find this comments useful for the improvement of the article.
Author Response
Dear Reviewer,
Thank you very much for your comments!
We have listened to your opinions carefully and made corresponding modifications to the manuscript. We also wrote an 'Author's reply' for your references, please see the attachment.
Kind Regards,
Meiling Han

Round 2
Reviewer 1 Report
Dear Authors,
Thank you very much for the new paper, this version has improved significantly and is way better in all aspects. Really interesting research!